

# Threshold in orbital forcing for Saharan greening lowers with rising levels of greenhouse gases

Mateo Duque-Villegas[1,2], Martin Claussen[1,3], Victor Brovkin[1], and Thomas Kleinen[1]

[1]Max Planck Institute for Meteorology, Hamburg, Germany
[2]International Max Planck Research School on Earth System Modelling, Hamburg, Germany
[3]Center for Earth System Research and Sustainability (CEN), Universität Hamburg, Hamburg, Germany

**Correspondence:** Mateo Duque-Villegas (mateo.duque@mpimet.mpg.de)

**Abstract.** Numerous climate archives reveal alternating arid and humid conditions in North Africa during the last several million years. Most likely the dry phases resembled current hyper-arid landscapes, whereas the wet phases known as African Humid Periods (AHPs) sustained much more surface water and greater vegetated areas that "greened" a large part of the Sahara region. Previous analyses of sediment cores from the Mediterranean Sea showed the last five AHPs differed in strength,

duration and rate of change. To understand the causes of such differences we perform transient simulations of the past 190,000 years with Earth system model of intermediate complexity CLIMBER-2. We analyse amplitude and rate of change of the modelled AHPs responses to changes in orbital parameters, greenhouse gases (GHGs) and ice sheets. In agreement with estimates from Mediterranean sapropels, we find the model predicts a threshold in orbital forcing for Sahara greening and occurrence of AHPs. Maximum rates of change in simulated vegetation extent at AHP onset and termination correlate well

with the rate of change of the orbital forcing. As suggested by available data for the Holocene AHP, the onset of modelled AHPs happens usually faster than termination. A factor separation analysis confirms the dominant role of the orbital forcing in driving the amplitude of precipitation and vegetation extent for past AHPs. Forcing due to changes in GHGs and ice sheets is only of secondary importance, with a small contribution from synergies with the orbital forcing. Via the factor separation we detect that the threshold in orbital forcing for AHP onset varies with GHGs levels. To explore the implication of our finding

from the palaeoclimate simulations for the AHPs that might occur in a greenhouse gas-induced warmer climate, we extend the palaeoclimate simulations into the future. For the next 100,000 years the variations in orbital forcing will be smaller than during the last hundred millennia, and the insolation threshold for the onset of late Quaternary AHPs will not be crossed. However, with higher GHGs concentrations the predicted threshold drops considerably. Thereby, the occurrence of AHPs in upcoming millennia appears to crucially depend on future concentrations of GHGs.

## 1 Introduction

Extensive evidence from geological records indicates that the landscape across North Africa changed repeatedly back and forth from wet to dry conditions during the late Quaternary (Larrasoaña et al., 2013; Grant et al., 2017). Wet phases termed African Humid Periods (AHPs) were intervals with increased rainfall, abundant lakes and rivers, as well as extended vegetation cover that "greened" large parts of the Sahara (deMenocal et al., 2000). Environmental shifts occurred at millennial timescale,



primarily in response to variations in the Earth's orbit, which altered the seasonal radiation budget and led to distinct regional circulation patterns and atmospheric moisture transports (Kutzbach, 1981). This link between orbital configuration and climate of North Africa is noticeable in proxy data (Lourens et al., 2001) and is supported by computer simulations (Tuenter et al., 2003). Orbitally induced changes in regional circulation were amplified by several feedback processes related mainly to surface properties such as sea temperature and land cover (Claussen et al., 2017; Pausata et al., 2020). Despite current knowledge about
these key mechanisms, inconsistencies between proxy data and simulations suggest there are still gaps in understanding of AHP dynamics (Braconnot et al., 2012).

    Much of what is known about AHPs stems out of the study of the last event during the Holocene epoch (Tierney et al., 2017). The relatively vast amount of available proxy data that cover this epoch has allowed for extensive data analyses (e.g., Bartlein et al., 2011; Lézine et al., 2011; Shanahan et al., 2015) and numerical simulations of its AHP (e.g., Jungandreas et al.,
2021; Cheddadi et al., 2021; Dallmeyer et al., 2020; Chandan and Peltier, 2020). Yet the latest sediment records from the Mediterranean Sea have highlighted the diversity in intensity of earlier AHPs (Blanchet et al., 2021; Ehrmann and Schmiedl, 2021), reaching as far back in time as Marine Isotope Stage (MIS) 6 about $190,000$ years ago (190 ka). The motivation behind this study lies in understanding the causes behind the different intensities. Although regional climate likely responded in a similar way for all previous AHPs, paced by orbital variations described in current theory (Claussen et al., 2017), an additional
complication emerges from the added effects of simultaneous changes in greenhouse gases (GHGs) and the waxing and waning of ice sheets. The two factors should act as additional forcing on the AHP response, considering that North African climate responds to changes in both of them (Claussen et al., 2003; Marzin et al., 2013), and assuming the region in turn has a negligibly small effect on these global climate drivers. Therefore we examine the changes in orbital, GHGs and ice sheets forcings in order to understand how the past five AHPs differed.

Previous modelling studies investigated the AHP response under these forcings separately, through sensitivity experiments that showed individually their first-order effects. For instance, Tuenter et al. (2003) described the consequences of changes in orbital parameters, while Claussen et al. (2003) focused on GHGs and Marzin et al. (2013) on the ice sheets. However, considering separately each forcing prevents simulation of synergical or joint effects amongst them, and therefore a direct comparison of their individual impact on AHP response is limited. The transient experiments in Weber and Tuenter (2011),
Kutzbach et al. (2020) and Blanchet et al. (2021) included all three forcing factors and showed the minor role the forcing from GHGs and ice sheets plays in setting the strength of AHPs. Nonetheless, these analyses focused more on the simulated response of North Africa (and how it compares with proxies) than on the evolution or absolute values of the forcings. The novelty of this work lies in our attention to the multiple forcings involved, since we look at their rates of change, threshold values and correlations, and we present the first factor separation analyses for these forcings in North Africa. Additionally we
use future estimates of these forcings to simulate potential future AHP responses and assess how far our lessons from past regional climate change can take us into the future.

    Our main goal is to investigate how much each forcing mechanism, by itself and in synergy with the others, contributes to the different AHPs intensities seen in proxy records. We use model CLIMBER-2 (Petoukhov et al., 2000; Ganopolski et al.,





2001) of intermediate complexity (Claussen et al., 2002), to study the climate response of North Africa to changes in orbital
parameters, GHGs radiative forcing and extent of ice sheets. We run an ensemble of transient global climate simulations for
the last 190 millennia (190 kyr) and study onset and termination dynamics of past AHPs. Using the factor separation method
of Stein and Alpert (1993) we quantify individual and synergical contributions of every forcing to the magnitude of an AHP.
Our findings from the past humid periods are also useful when we extend experiments for a 100 kyr into the future and assess
potential consequences of changes in GHGs in relation to development of future AHPs.

## 2 Methods

### 2.1 Model description

CLIMBER-2 incorporates a 2.5 dimensional statistical–dynamical atmosphere model of coarse horizontal resolution of about
$10°$ latitude and $51°$ longitude, and a parameterised vertical structure assuming universal profiles of temperature, humidity
and meridional circulation (Petoukhov et al., 2000). The atmosphere component is coupled to a zonally averaged and multi-
basin dynamic ocean model based on that of Stocker et al. (1992), with meridional resolution of $2.5°$ latitude and 20 vertical
levels. Connected to these components the model also includes a one layer thermodynamic sea ice model with horizontal ice
transport (Ganopolski and Rahmstorf, 2001), the three-dimensional polythermal ice sheet model SICOPOLIS (Greve, 1997),
the dynamic global vegetation model VECODE (Brovkin et al., 1997), a dynamic global carbon cycle model with land and
ocean biogeochemistry (Brovkin et al., 2002, 2007; Ganopolski and Brovkin, 2017), and modules for aeolian dust effects
(Bauer and Ganopolski, 2010; Ganopolski et al., 2010).

The model has a low computational cost that enables simulations over long timescales, when multiple components of the
climate system can interact. Despite its coarse spatial resolution and simplifications the model captures well past climate
changes (Claussen et al., 1999a), as well as the aggregated large–scale features of modern climate (Ganopolski et al., 1998).
Its response to changes in boundary conditions and climate forcings is comparable to that of more comprehensive models
(Ganopolski et al., 2001), and it successfully simulates glacial–interglacial cycles (Ganopolski et al., 2010; Ganopolski and
Brovkin, 2017). In fact, CLIMBER-2 is currently the only geographically explicit model which can be used for an ensemble of
simulations of glacial–interglacial cycles. For the specific case of North Africa CLIMBER-2 has already been used to study its
climate on multi-millennia timescales with a favourable performance (e.g., Claussen et al., 1999b, 2003; Tuenter et al., 2005;
Tjallingii et al., 2008).

### 2.2 Experiments of past AHPs

We simulate the coupled atmosphere, ocean, sea ice, land surface and vegetation dynamics for the past 190 kyr. The poly-
thermal ice sheets model and global carbon cycle model components are not employed since we prescribe ice volume and
atmospheric GHGs levels using available data. Simulations start from an equilibrium state attained after a 5 kyr simulation
that maintained parameters set at 190 ka values. We perform 15 transient simulations prescribing possible combinations of





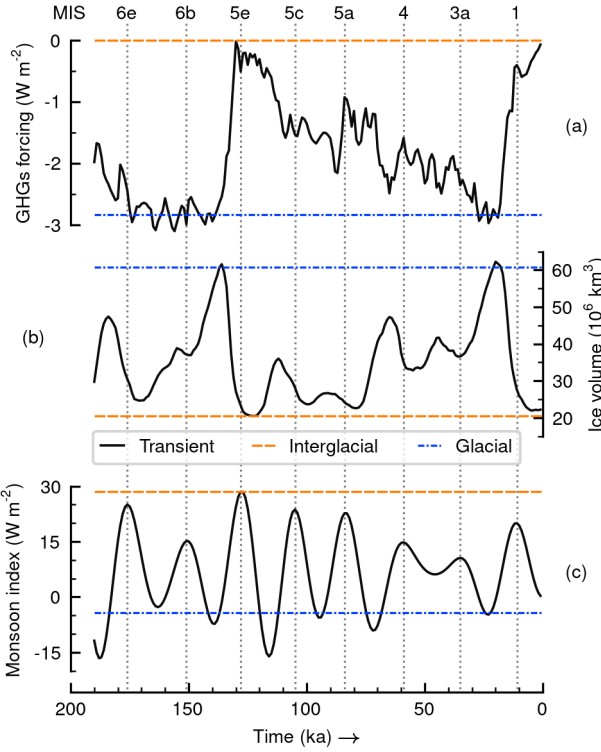

**Figure 1.** Prescribed forcing parameters in simulations for (a) GHGs radiative forcing, (b) ice sheets and (c) orbital forcing. Transient series come from Antarctic ice cores, modelled ice sheets data and orbital theory, respectively. Horizontal lines are cases when forcings are fixed to interglacial or glacial reference values. Vertical lines indicate maxima in the monsoon index of tropical insolation and are labelled using MIS names on top (see Appendix A about notation).

climate forcing factors: GHGs levels, ice sheets extent and orbital parameters. Forcings are prescribed either with a transient series from past evidence or with a single reference value as shown in Fig. 1. Reference values are taken from the transient series close to the Eemian interglacial state at about $125\,\mathrm{ka}$ and close to the Last Glacial Maximum (LGM) at about $21\,\mathrm{ka}$. The GHGs radiative forcing in Fig. 1a comes from Ganopolski and Calov (2011) and it considers variations in gases $CO_2$, $CH_4$ and $N_2O$ that are in line with Antarctic ice cores (Petit et al., 1999; EPICA Community Members, 2004). Ice sheets data are

model output from a simulation also in Ganopolski and Calov (2011) which agrees with sea level changes in Waelbroeck et al. (2002) and the reconstructions of Peltier (1994). Ice sheets are represented by a spatially distributed transient series varying mostly in the Northern Hemisphere, albeit we show only the global cumulative ice volume (Fig. 1b). Earth's orbital parameters (precession, obliquity and eccentricity) vary according to Berger (1978). We show their secular variations using the monsoon index defined by Rossignol-Strick (1983), which measures an insolation gradient in the tropics relevant to monsoon systems

and AHPs (Fig. 1c). In the text and in Fig. 1 we label peaks in the monsoon index using MIS names to ease location of time



**Table 1.** Experiments and forcing settings. Entries in "Field" column indicate how a climatic variable taken from an experiment is used in the factor separation method (see Appendix B with equations). Transient series are those in Fig. 1. The future set is not part of the separation method. 0 ka is present-day and AP is after present-day.

| Name | Time | Field | Description | GHGs radiative forcing (W m$^{-2}$) | Ice volume ($10^6$ km$^3$) | Monsoon index (W m$^{-2}$) |
|---|---|---|---|---|---|---|
| E0 | 190 ka–0 ka | $f_0$ | Control experiment. | Transient | Transient | Transient |
| EI1 | | $f_{I, GHG}$ | | 0.0 | Transient | Transient |
| EI2 | | $f_{I, Ice}$ | | Transient | 20.5 | Transient |
| EI3 | | $f_{I, Orbital}$ | | Transient | Transient | 28.5 |
| EI4 | 190 ka–0 ka | $f_{I, GHG + Ice}$ | Interglacial separation set. | 0.0 | 20.5 | Transient |
| EI5 | | $f_{I, GHG + Orbital}$ | | 0.0 | Transient | 28.5 |
| EI6 | | $f_{I, Ice + Orbital}$ | | Transient | 20.5 | 28.5 |
| EI7 | | $f_{I, GHG + Ice + Orbital}$ | | 0.0 | 20.5 | 28.5 |
| EG1 | | $f_{G, GHG}$ | | -2.8 | Transient | Transient |
| EG2 | | $f_{G, Ice}$ | | Transient | 60.7 | Transient |
| EG3 | | $f_{G, Orbital}$ | | Transient | Transient | −4.2 |
| EG4 | 190 ka–0 ka | $f_{G, GHG + Ice}$ | Glacial separation set. | −2.8 | 60.7 | Transient |
| EG5 | | $f_{G, GHG + Orbital}$ | | −2.8 | Transient | −4.2 |
| EG6 | | $f_{G, Ice + Orbital}$ | | Transient | 60.7 | −4.2 |
| EG7 | | $f_{G, GHG + Ice + Orbital}$ | | −2.8 | 60.7 | −4.2 |
| F0 | | – | | 0 Gt C scenario | 22.2 | Berger (1978) |
| F1 | | – | | 1000 Gt C scenario | 22.2 | Berger (1978) |
| F2 | | – | | 2000 Gt C scenario | 22.2 | Berger (1978) |
| F3 | 0 ka–100 kyr AP | – | Future set. | 3000 Gt C scenario | 22.2 | Berger (1978) |
| F4 | | – | | 4000 Gt C scenario | 22.2 | Berger (1978) |
| F5 | | – | | 5000 Gt C scenario | 22.2 | Berger (1978) |

slices when AHPs usually occur. Details about forcings, computation of the monsoon index and MIS nomenclature are given in Appendix A.

Experiments of past AHPs and their forcing setup combinations follow the factor separation method of Stein and Alpert (1993) for three factors: (1) GHGs radiative forcing, (2) ice sheets extent and (3) orbital parameters. Using this method it
is possible to estimate individual and synergical contributions of the forcing factors to a simulation outcome (or predicted climatic field). We are interested in knowing how much the forcings contribute to the simulation of AHPs. To implement the method a specific simulation target must be chosen in order to calculate deviations from the baseline or control state (i.e., the

factor separation depends on the target or point of view). We choose two simulation targets with opposing global climates and AHP situations: the Eemian interglacial around 125 ka (i.e., warm global climate) with a strong AHP and the LGM around
21 ka (i.e., cold global climate) with no AHP. Experiments and forcings combinations are shown in Table 1. Only in the control experiment E0 all forcings vary realistically as in the transient series obtained from past data (see Fig. 1). For the rest of experiments we have two sets of simulations using the interglacial or glacial reference values shown in Table 1, with all possible "fixed-or-transient" combinations in each set. For instance, simulation EI4 has both GHGs forcing and ice sheets fixed at the interglacial reference values for the entire 190 kyr run (i.e., they are kept constant at interglacial levels). Likewise for
EG4 but instead the fixed values are the glacial levels. Experiments EI7 and EG7 have the three forcings fixed at their respective reference points. Details and equations of the separation analyses are given in Appendix B.

## 2.3 Experiments of future AHPs

The low computational cost of the model and information available about future changes in the forcings enable us to also look into potential future climate change in North Africa. We are interested in knowing how much our findings from the past can
inform the future. Additional simulations start from present-day conditions in the control experiment E0 and cover the next 100 millennia after present (kyr AP). Projections of GHGs radiative forcing are based on the $CO_2$ emissions scenarios of Archer and Brovkin (2008), which differ by the cumulative amount of carbon released to the atmosphere in units of gigatonnes of carbon (Gt C). Five experiments include a scenario of null emissions since preindustrial conditions (0 Gt C), a "moderate" emissions scenario with a release of 1000 Gt C, a "large" emissions scenario with a release of 5000 Gt C, and the intermediate
cases of 2000, 3000 and 4000 Gt C. In these scenarios 90 % of the emissions are assumed to occur within the first few centuries of simulation, meaning there is an early peak of atmospheric concentration of $CO_2$ that subsequently decays to an equilibrium value (higher than preindustrial). These experiments are also shown in Table 1. For these simulations we ignore the radiative forcing effect of other GHGs. The orbital forcing is considered to continue changing following Berger (1978). We keep the ice sheets fixed at the preindustrial setting, neglecting the effects of a potential complete deglaciation or a new glacial inception.
The latter is justified since a large ice sheet should not emerge within the next 100 kyr AP, even for the moderate emissions scenario (Ganopolski et al., 2016).

## 3 Results

We study model output of century-mean values of near-surface air temperature, daily precipitation and fractional vegetation coverage in the Sahara region. In the model the Sahara is located in one grid box spanning approximately 20–30° N and 15° W–
50° E. Temperature and precipitation are resolved daily in the model, hence we include seasonal averages for them, whereas vegetation is resolved annually, thus only annual mean values are reported. Summer refers to the June–July–August average, while winter to the December–January–February average.





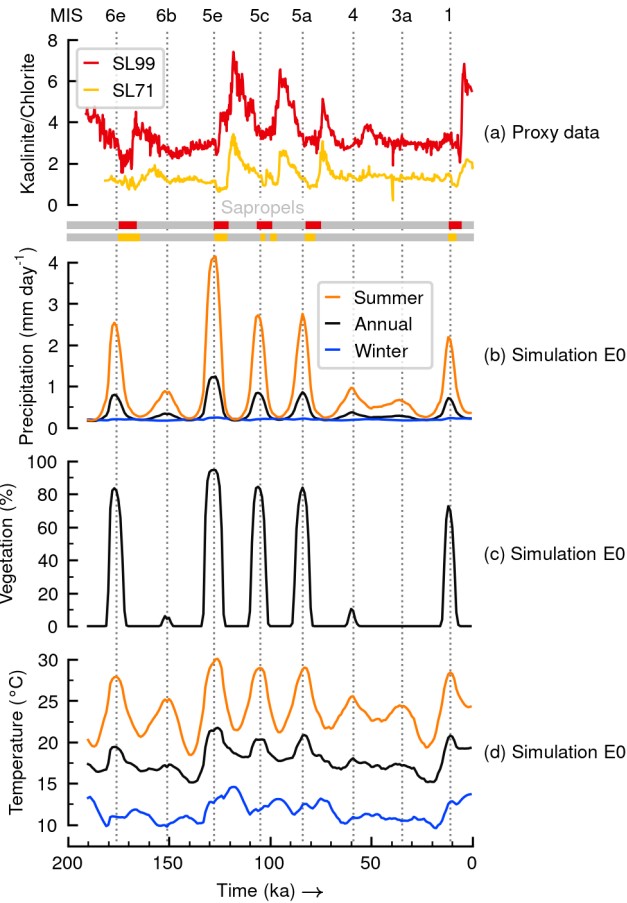

**Figure 2.** Comparison of (a) proxy data with modelling results in Sahara for (b) daily precipitation, (c) vegetation fraction and (d) near-surface temperature in control simulation E0. Proxy data are clay minerals ratios and sapropel layers from two sediment cores from Eastern Mediterranean described in Ehrmann and Schmiedl (2021). Vertical lines indicate maxima in the monsoon index of tropical insolation and are labelled using MIS names on top.

## 3.1 Simulation of past AHPs

Results of the control simulation E0 are shown in Fig. 2 alongside proxy data from sediment cores from the Eastern Mediter-
ranean Sea (Ehrmann and Schmiedl, 2021). The proxies in Fig. 2a show dust pulses of clay minerals found always following
a sapropel layer in the sediments. Sapropel layers indicate past instances of AHPs, while the amplitudes of the dust pulses
are linked to the strength of the hydrological cycle during their preceding AHPs. The comparison between the proxy data and
our simulations can only be qualitative in nature, since we do not model either dust transport into the Mediterranean Sea nor
sapropel formation. It is remarkable, however, that the control simulation E0 (Fig. 2b–d) shows large peaks close in time with
the sapropel layers in the proxies. Also both in simulation E0 and the proxies the strongest signals happen during MIS 5e.





**Table 2.** Results in Sahara of annual mean values in control experiment E0 taken near maxima of the monsoon index of tropical insolation. Sapropel layers are those in Fig. 2a. Values in parentheses are winter and summer means.

| MIS | Time (ka) | Forcing values | | | Simulated responses | | | |
| | | GHGs forcing $(\mathrm{W\,m^{-2}})$ | Ice volume $(10^6\,\mathrm{km^3})$ | Monsoon index $(\mathrm{W\,m^{-2}})$ | Temperature $(^\circ\mathrm{C})$ | Precipitation $(\mathrm{mm\,day^{-1}})$ | Vegetation $(\%)$ | Sapropel layer |
|---|---|---|---|---|---|---|---|---|
| 6e | 176 | −2.4 | 30.5 | 25.0 | 19.4 (11.0, 27.9) | 0.8 (0.2, 2.5) | 83.5 | Yes |
| 6b | 151 | −3.0 | 37.1 | 15.1 | 17.3 (10.1, 25.2) | 0.3 (0.2, 0.9) | 6.1 | No |
| 5e | 128 | −0.5 | 23.3 | 28.5 | 21.6 (13.0, 30.0) | 1.2 (0.2, 4.1) | 94.5 | Yes |
| 5c | 105 | −1.5 | 28.2 | 23.6 | 20.3 (12.0, 29.0) | 0.8 (0.2, 2.7) | 84.4 | Yes |
| 5a | 84 | −0.9 | 24.1 | 22.7 | 20.8 (12.5, 29.0) | 0.9 (0.2, 2.8) | 83.8 | Yes |
| 4 | 59 | −1.6 | 35.1 | 14.8 | 18.0 (10.9, 25.6) | 0.4 (0.2, 1.0) | 10.4 | No |
| 3a | 35 | −2.4 | 36.6 | 10.5 | 17.3 (10.6, 24.4) | 0.3 (0.2, 0.7) | 0.0 | No |
| 1 | 11 | −0.4 | 27.5 | 20.0 | 20.8 (12.8, 28.4) | 0.7 (0.2, 2.2) | 72.6 | Yes |

The simulation E0 is in closer agreement with the data from core SL71, where the pulses after MIS 5c and 5a have similar magnitude.

Simulation E0 shows peak values of precipitation, vegetation fraction and temperature in the Sahara region coincide with peaks in the monsoon index during the past 190 kyr. The peak values are summarised in Table 2. In the case of precipitation (Fig. 2b) and temperature (Fig. 2d) this is most conspicuous in the summer seasonal means, with large peaks similarly as evident as those in the vegetation fraction (Fig. 2c). The strongest peak values occur during the last two interglacial periods at stages MIS 5e and 1, as well as during stages MIS 6e, 5c and 5a. Only these five MIS periods have a vegetation fraction above 70 %. The vegetation peaks happen during warmer periods when annual mean daily precipitation is greater than $0.7\,\mathrm{mm\,day^{-1}}$. This can accumulate to more than $250\,\mathrm{mm\,year^{-1}}$, well above the minimum limit estimated to support perennial grasslands and savanna biome types in North Africa (Larrasoaña et al., 2013). Although this precipitation value is low compared to estimates from proxy data (Braconnot et al., 2012), any amount over $200\,\mathrm{mm\,year^{-1}}$ is still distinctly greater than present-day values across the region (Bartlein et al., 2011), and is within the variability range of more sophisticated climate models for the last AHP during the Holocene (Harrison et al., 2015). Therefore we consider these peak times to be simulated past AHPs with CLIMBER-2. The oldest one at MIS 6e was also found by Tuenter et al. (2005), and those at MIS 5c, 5a and 1, agree with the results of Tjallingii et al. (2008). Measuring from the moment the modelled vegetation starts growing until the moment it vanishes (Fig. 2c), the AHPs last on average 10.8 kyr, with the one at MIS 5e being the longest (12 kyr) and the one at MIS 1 the shortest (9 kyr).





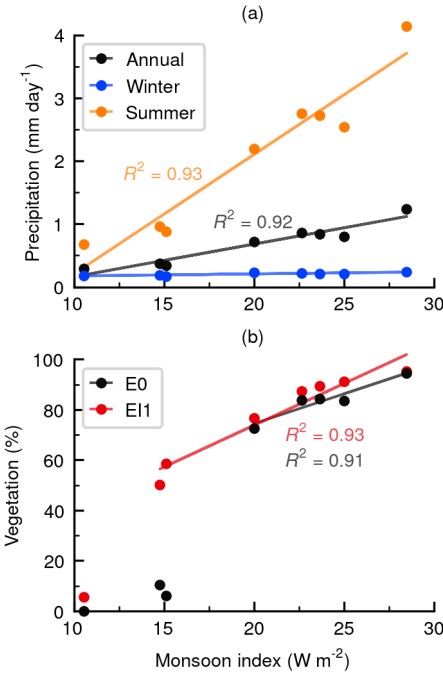

**Figure 3.** Correlation of orbital forcing with modelled responses in Sahara in control simulation E0 for mean values of (a) daily precipitation and (b) vegetation fraction. In (b) we also include the modelled vegetation from experiment EI1 with fixed interglacial GHGs.

### 3.2 Correlation of forcings and AHPs

A first approximation to study the effects of the forcings on the modelled AHP response uses a simple linear regression
analysis. With the values from Table 2 we plot paired combinations of forcings and response variables in phase diagrams and fit regression lines with the ordinary least-squares method and use the statistic $R^2$ and the $p$-value of the F-statistic as goodness-of-fit estimates. The forcings are included in the regression analysis (besides the modelled response) because even though we know that GHGs levels and ice sheets are not independent from the orbital forcing, we do not know exactly how they relate to each other during AHPs. We also reckon GHGs and ice sheets are global features while the monsoon index is a
regional insolation gradient. We find there is a strong correlation ($R^2 = 0.75$, $p$-value $= 0.006$) between the average ice sheets volume and the monsoon index, with the highest monsoon index values having the least ice volume. In contrast, we see a weak correlation ($R^2 = 0.27$, $p$-value $= 0.19$) between the GHGs radiative forcing and the tropical insolation gradient imposed by the orbital forcing.

The modelled response correlates strongly with the monsoon index as shown in Fig. 3. For precipitation (Fig. 3a) we observe
a positive linear relationship for summer seasonal ($R^2 = 0.93$, $p$-value $= 0.0001$) and annual ($R^2 = 0.92$, $p$-value $= 0.0001$) means, therefore higher monsoon index values yield higher precipitation in the Sahara region. Winter precipitation being almost a flat line shows this modelled variable in Sahara does not depend on the monsoon index. In the case of vegetation (Fig. 3b)

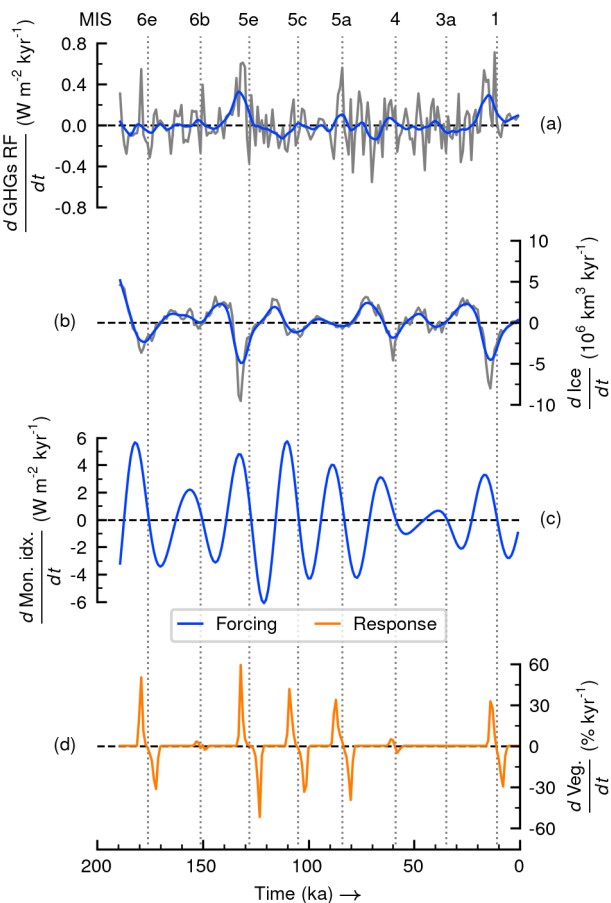

**Figure 4.** Rates of change of forcings and modelled vegetation in control experiment E0: (a) GHGs Radiative Forcing (RF), (b) global ice volume, (c) monsoon index and (d) vegetation fraction in Sahara. GHGs and ice volume series are smoothed using a Locally Weighted Scatterplot Smoothing (LOWESS) filter of 15 kyr. Vertical lines show maxima of monsoon index of tropical insolation and are labelled using MIS names on top.

we see non-linear behaviour with an abrupt large jump from low (about 10 %) to high (about 70 %) vegetation coverage within a small range of variation of the monsoon index from around 15 to 20 $\mathrm{W\,m^{-2}}$. Only for vegetation a signal change-point
analysis yields one clear change-point at 20 $\mathrm{W\,m^{-2}}$ (not shown; Killick et al., 2012). Vegetation values occurring past the threshold range have a strong correlation ($R^2 = 0.91$, $p$-value $= 0.012$) with the monsoon index. We also include in Fig. 3b the vegetation response of sensitivity simulation EI1 (with interglacial GHGs forcing) because it shows that the insolation threshold might be sensitive to the GHGs forcing. With interglacial GHGs radiative forcing (experiment EI1) the threshold decreases below 15 $\mathrm{W\,m^{-2}}$ and the correlation past this threshold between vegetation and monsoon index is still strong
($R^2 = 0.93$, $p$-value $= 0.0005$).





**Table 3.** Rates of change of forcings and modelled vegetation in control simulation E0 at times of peak rates in vegetation response. Only shown are MIS stages with an AHP response. For GHGs Radiative Forcing (RF) and global ice volume the values are from the smoothed series in Fig. 4.

| | | | *Forcings* | | *Response* |
|---|---|---|---|---|---|
| MIS | Time (ka) | $\dfrac{d\text{GHGs RF}}{dt}$ $(\text{W m}^{-2}\,\text{kyr}^{-1})$ | $\dfrac{d\text{Ice}}{dt}$ $(10^6\,\text{km}^3\,\text{kyr}^{-1})$ | $\dfrac{d\text{Mon. idx.}}{dt}$ $(\text{W m}^{-2}\,\text{kyr}^{-1})$ | $\dfrac{d\text{Veg.}}{dt}$ $(\%\,\text{kyr}^{-1})$ |
| | | | *Inception AHP* | | |
| 6e | 179 | 0.0 | −2.3 | 4.2 | 50.1 |
| 5e | 132 | 0.3 | −4.9 | 4.8 | 59.2 |
| 5c | 109 | −0.1 | −0.8 | 5.4 | 41.6 |
| 5a | 87 | 0.1 | −0.3 | 3.6 | 33.7 |
| 1 | 14 | 0.3 | −4.6 | 2.5 | 32.6 |
| | | | *Termination AHP* | | |
| 6e | 172 | 0.0 | −0.8 | −3.0 | −31.6 |
| 5e | 123 | 0.0 | −0.1 | −5.3 | −52.1 |
| 5c | 102 | 0.0 | −0.8 | −3.2 | −33.7 |
| 5a | 80 | 0.0 | −0.1 | −3.1 | −39.7 |
| 1 | 8 | 0.0 | −1.2 | −2.1 | −29.8 |

## 3.3 Speed of change of AHPs

Proxy data indicates that past AHPs were different not only in magnitude, but also in their rates of change (Ehrmann et al., 2017). Figure 4 shows finite differences with respect to time in the forcings and modelled vegetation response in the control experiment E0. The radiative forcing from the GHGs increases quickly towards positive values just before the two interglacial states at MIS 5e and 1 (Fig. 4a). Everywhere else GHGs changes happen relatively slowly and erratically. Ice volume changes happen more smoothly. There is a strong and quick reduction of ice sheets before interglacial states at MIS 5e and 1 (Fig. 4b). Then milder in strength follow the ice reduction speeds just before stages MIS 6e, 5c and 4. During MIS 5a the ice sheets do not change as much prior to the monsoon index peak, however, soon after it there is a relatively quick growth of ice sheets. For the monsoon index we notice that extreme speeds of growth and decline occur around the vertical lines (Fig. 4c), where the maximum values happen (inflection points). The fastest increases in monsoon index occur at stages MIS 6e and 5c. Then follow closely stages at MIS 5e and 5a. Remaining stages have relatively weaker speeds of change for this variable. The fastest reduction in the monsoon index happens after stage MIS 5e.

Figure 4 shows that the changes in the vegetation response do not follow linearly one single forcing. Nevertheless, they share the inflection points with the monsoon index rate of change, having both maximum growth and decline speeds around



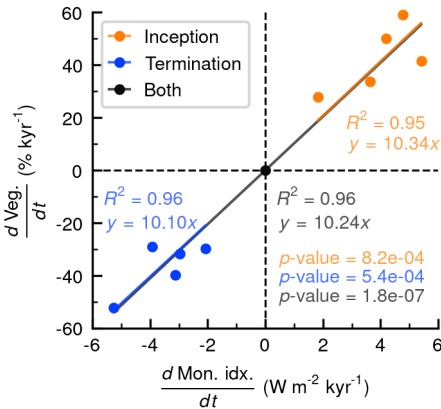

**Figure 5.** Correlation of peak rates of change in vegetation response in Sahara versus rates of change in monsoon index during AHP inception and termination in control simulation E0.

the vertical lines. Vegetation growth maxima occur always about $2.0\,\mathrm{kyr}$ after the monsoon index peak rates of change, while maximum vegetation decline rate occurs on average $2.4\,\mathrm{kyr}$ before the minimum peaks in monsoon index speed of change (Fig. 4d). Peak rates of positive and negative change are summarised in Table 3. The fastest vegetation growth rates occur at stages MIS 6e and 5e, while the slowest happens at MIS 1. The fastest decline rates in vegetation are seen after MIS 5e and 5a. In general, we see that the maximum vegetation growth speeds are faster than subsequent declining rates, the clearest example being MIS 6e. Only at MIS 5a the opposite is true.

Table 3 shows that the strongest vegetation growth rate during the AHP inception in MIS 5e occurs simultaneously with the strongest increase rate in GHGs radiative forcing and the strongest reduction rate in ice volume, in spite of the monsoon index increase rate not being the largest. Most likely during AHP inception in MIS 1, the increasing rate of GHGs radiative forcing and the strong reduction of ice sheets compensated for the slow rate of change in monsoon index. In the case of terminations of AHPs, the leading factor contributing to the speed of vegetation decline is the change in the monsoon index. Figure 5 shows the relationship between these two rates of change during inception and termination of AHPs. We find a strong correlation ($R^2 = 0.96$, $p$-value $< 0.0001$) between the speed of change in the monsoon index and the peak speed of growth and decline of vegetation in the model. The larger the positive (negative) monsoon index change rate, the larger the growth (decline) rate in vegetation during AHP inception (termination). The residual errors in the regression lines presumably include the effects of the changes in the other forcings.

### 3.4 Factor separation analyses

We suspect that differences in intensity amongst AHPs should be mainly determined by differences in the state of GHGs, ice sheets and orbital configuration at the time they occur. However, we do not know how much each of the forcings is affecting the AHPs intensities. To look into this we perform simulations according to the factor separation method of Stein and Alpert





(1993) and implement its equations (see Appendix B) using the model output for annual means of temperature, precipitation and vegetation fraction in the Sahara region. Because the separation method depends on the chosen simulation target we carry out two analyses: one of them isolates the contributions of the forcing factors to attain an Eemian-like strong AHP response (i.e., interglacial perspective), while another one does it for an LGM-like non-AHP response (i.e., glacial perspective).

Both target responses for the two separation analyses are represented in simulations EI7 and EG7. Simulation EI7 from

the interglacial perspective has all three forcing factors fixed at interglacial conditions (using the Eemian as reference). In this simulation modelled Saharan temperature stays for all times at 21.8 °C, precipitation at 1.3 mm day$^{-1}$ and vegetation fraction at 95.8 %. These values resemble those around the Eemian peak (about 125 ka) in the control simulation E0 (see Fig. 2). Simulation EG7 from the glacial perspective has forcings fixed at glacial conditions (using LGM as reference). In this simulation modelled Saharan temperature remains for all times at 15.2 °C, precipitation at 0.2 mm day$^{-1}$ and vegetation

fraction at 0.0 %. These values resemble those around the LGM (about 21 ka) in the control simulation E0 (see Fig. 2). Therefore, the interglacial perspective experiment EI7 keeps a permanent strong Eemian-like AHP for the entire simulation, whereas the glacial EG7 keeps none. Through the factor separation method we then estimate how much each of the forcings is contributing at all times to the differences between the control experiment E0 and targets EI7 (strong AHP) and EG7 (no AHP).

Results of the separation analysis from the interglacial perspective are shown in Fig. 6 for AHP temperature, precipitation and vegetation fraction. With the forcing factors fixed at interglacial levels the modelled response in EI7 (Eemian-like AHP) deviates always positively from the control experiment E0 for the three climatic variables. This is as expected since the resulting values of EI7 in the previous paragraph are higher than those in the control E0 at all times (only Eemian in E0 is close to EI7). The relevant outcome of the separation method is in the colours in Fig. 6, which show how the forcings contribute to the

EI7 − E0 difference. For AHP temperature (Fig. 6a) the individual contribution of each forcing is directly proportional to the change in the forcing. To see this it is necessary to compare differences between the control forcing values in the transient series in Fig. 1 and the interglacial reference level in that same figure. For instance, at stage MIS 6b the largest contribution to temperature change comes from the GHGs radiative forcing change (Fig. 6a), since that is the forcing with the largest difference between the transient series and the interglacial horizontal line (see Fig. 1a). For the AHP temperature response the synergistic

effects are small in comparison to the individual contributions. Averaging the contributions to AHP temperature for the whole time series in Fig. 6a yields that GHGs add about 1.6 °C (41.1 % of change from E0), ice sheets 0.8 °C (19.9 % of change from E0) and the orbital forcing 1.8 °C (45.9 % of change from E0). The synergy between GHGs and the orbital forcing is the one with the largest effect adding in average about −0.2 °C (−4.9 % of change from E0).

The AHP precipitation case in the separation analysis is different to the temperature one. Figure 6b shows that the AHP

precipitation response largely depends on the orbital forcing. Not only the individual contribution of this forcing is the largest for all times, but also its synergies with the other two forcings have relatively large effects. The individual contribution of the GHGs radiative forcing to changes in AHP precipitation is small. The individual effects of ice sheets on AHP precipitation are negligible as well. No synergistic effects exist between the ice sheets forcing and the GHGs radiative forcing. Averaging





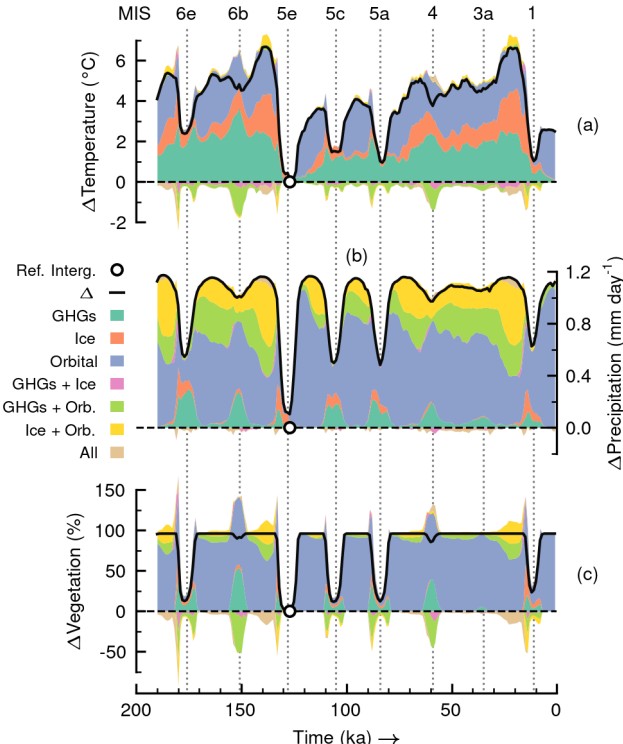

**Figure 6.** Factor separation analysis in Sahara from interglacial perspective for annual means of (a) near-surface temperature, (b) daily precipitation and (c) vegetation fraction. Colours indicate contributions from individual forcings and related synergies to the total deviation in EI7 (Eemian-like AHP) from the control experiment ($\Delta = $ EI7 $-$ E0). A marker shows the location of the reference interglacial state, where the difference between EI7 and E0 is minimum. Vertical lines indicate maxima of monsoon index of tropical insolation and are labelled using MIS names on top.

the contributions to AHP precipitation for the whole time series in Fig. 6b we obtain that GHGs add about $0.1\,\mathrm{mm\,day^{-1}}$

(6.7 % of change from E0), ice sheets close to $0.0\,\mathrm{mm\,day^{-1}}$ (2.2 % of change from E0) and the orbital forcing about $0.6\,\mathrm{mm\,day^{-1}}$ (59.3 % of change from E0). Both synergies of the orbital forcing with GHGs and ice sheets round up each one to $0.2\,\mathrm{mm\,day^{-1}}$ (each about 15 % of change from E0). This means the effects of the orbital forcing and these two synergies with GHGs and ice sheets amount to about 90.9 % of the change from E0.

AHP vegetation results in Fig. 6c are similar to the AHP precipitation case, although the individual effect of the orbital

forcing is much more conspicuous (predominant blue colour). The negative synergistic effects seen in the vegetation changes are an artefact of the separation analysis method because vegetation is a bounded quantity. Consider, for example, the time around 150 ka during MIS 6b, when the Sahara is much greener, close to a 100 % vegetation cover, in simulation EI7 than in simulation E0 (Fig. 6c). The individual contribution of orbital forcing (blue colour) would make some 90 % of the Sahara green, and the GHGs forcing (muted green colour) some 50 %. If both forcings are active, then the pure contributions do not





add linearly, because the Sahara cannot be more than 100 % covered with vegetation. Therefore, the synergy between these forcings (light green colour) has to be negative. Average contributions to AHP vegetation for the whole series in Fig. 6c are about 5.8 % from GHGs (6.4 % of change from E0), 2.2 % from ice sheets (2.4 % of change from E0) and 72.6 % from the orbital forcing (79.5 % of change from E0). The synergy (when positive) between GHGs and orbital forcing adds in average about 4.0 % (4.4 % of change from E0) while the synergy (when positive) between ice sheets and orbital forcing about 5.3 % (5.8 % of change from E0). The orbital forcing and all of its synergies account for about 90.4 % of vegetation change from E0.

From the glacial perspective in the additional separation analysis we obtain qualitatively similar results (Fig. C1). In this case the deviations in EG7 from the control E0 are always negative for the three climatic variables and the effects of the ice sheets changes are larger than those of the GHGs. Nonetheless, the same behaviour for temperature is observed, with changes being proportional to the changes in the forcings. We also see the orbital forcing as the main contributor to the reductions in precipitation and vegetation that keep the non-AHP conditions during glacial times. From this perspective synergies are more difficult to assess because of the lower bounds in precipitation ($0 \ \mathrm{mm \ day^{-1}}$) and vegetation ($0$ %). Details of this separation analysis are given in Appendix C.

### 3.5 AHPs in scenarios of future climate change

When computing the monsoon index for the next $100 \ \mathrm{kyr \ AP}$ (millennia after present) we see that the onset threshold of past AHPs (between $15$–$20 \ \mathrm{W \ m^{-2}}$ in the model) will not be crossed within the next some $60 \ \mathrm{kyr}$ (Fig. 7a). The amplitude in the monsoon index (proxy for orbital forcing) is much smaller than that of past times (see Fig. 1) due to low eccentricity in the Earth's orbit. Only for the time around $66 \ \mathrm{kyr \ AP}$ the monsoon index approaches the onset threshold. From the sensitivity simulation EI1 in which the GHGs forcing is set to interglacial level, we already find the orbital threshold can change (see Fig. 3b). Hence we expect that in a climate with much stronger GHGs forcing (Fig. 7b) the threshold might decrease even further. The simulations F0–5 corroborate the lesson learnt from the study of the past (see Fig. 7c). Emissions scenarios in the simulations have a peak GHGs forcing at the start since they assume that $90$ % of carbon is released within the first centuries. As mentioned in Section 2.3, we assume that global ice volume stays close to its current low level and ignore the effects of a total deglaciation or a new glacial inception.

The modelled vegetation response in Fig. 7c shows the potential effects of increased GHGs radiative forcing for climate in the Sahara. The no-emissions scenario ($0 \ \mathrm{Gt \ C}$) predicts that no AHP should occur before the next $60 \ \mathrm{kyr \ AP}$. For this scenario only at $66 \ \mathrm{kyr \ AP}$ (named peak M4) the threshold of the monsoon index of tropical insolation is approached slightly at $16.5 \ \mathrm{W \ m^{-2}}$, and we see a vegetation response of about $54.1$ %. However, for the "moderate" emissions scenario ($1000 \ \mathrm{Gt \ C}$) we see already that additional vegetation responses begin to occur at earlier monsoon index peaks. These mild responses at other monsoon index peaks are amplified as the GHGs radiative forcings increase with the scenarios. For the "large" emissions scenario ($5000 \ \mathrm{Gt \ C}$) the response at $9 \ \mathrm{kyr \ AP}$ (named peak M1) with monsoon index of only $9.2 \ \mathrm{W \ m^{-2}}$ is already at about $89.9$ %, which is higher than that of the strongest monsoon index peak M4 for any scenario. Therefore we see that the effect of the large GHGs radiative forcing in the early $60 \ \mathrm{kyr \ AP}$ of simulation is compensating for the low monsoon index values




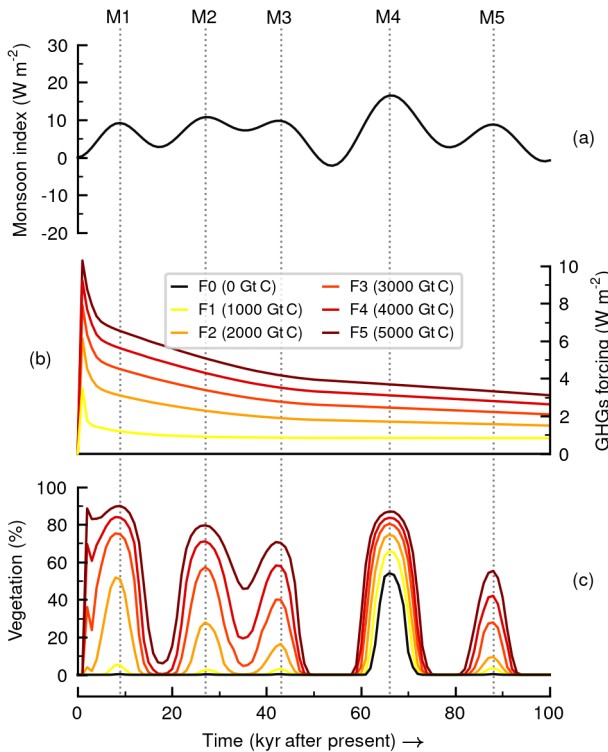

**Figure 7.** AHPs in future climate change scenarios for the next 100 kyr AP: (a) computed monsoon index changes, (b) scenarios of GHGs radiative forcing and (c) modelled vegetation fraction in Sahara. Vertical lines indicate future peaks in monsoon index of tropical insolation. Total cumulative emissions of carbon are shown in parentheses in the legend.

during this time. Nonetheless, this effect is only seen at the monsoon index peaks, therefore the orbital forcing is still the triggering mechanism for the AHP response.

## 4 Discussion

We initially assess the ability of model CLIMBER-2 to simulate past occurrences of AHPs during glacial–interglacial cycles. Our results agree with previous efforts which also found the model's performance adequate (e.g., Tuenter et al., 2005; Tjallingii et al., 2008). Magnitude and timing of the simulated AHPs compare favourably not only with the proxy data in Ehrmann and Schmiedl (2021), but also with the data and simulations of Blanchet et al. (2021) and Kutzbach et al. (2020). Although comparison between simulations is limited due to large differences in spatial resolution, one difference that might be discussed is the magnitude of the simulated Holocene AHP. In the case of Blanchet et al. (2021) the AHP during the Holocene is stronger than those at MIS 5c and 5a. Our simulations show the opposite, in agreement with the simulations in Kutzbach et al. (2020). To reconcile this we can look at the sediment records from the Mediterranean Sea discussed in Ehrmann and Schmiedl (2021) and





say that simulations of Saharan climate in CLIMBER-2 evolve like the proxy data from semi-distal sites off the North African
coastline, whereas those in Blanchet et al. (2021) resemble the data from proximal sites. Spatial heterogeneity of the changes
during AHPs hinders model consistency with all proxy data. Nevertheless our simulations show a reasonable representation of
the evolution of the climate in the Sahara during the last 190 kyr and this enables us to study the effects of forcing from GHGs,
ice sheets and orbital parameters.

Analysis of our control simulation reveals that the AHP response in CLIMBER-2 is highly correlated with the variations of
the monsoon index (proxy for orbital forcing based on a gradient in tropical insolation). This correlation was also found in the
data of Rossignol-Strick (1983) and Ehrmann et al. (2017). We find that there is a critical value of the monsoon index that must
be crossed to simulate Saharan greenings with the model. Past this threshold vegetation and precipitation feed back through
biogeophysical changes and the climatic response in Sahara is amplified resulting in an AHP (Brovkin et al., 1998; Claussen
et al., 1999b). Such non-linear responses from monsoon regions are also seen in proxy data (Rossignol-Strick, 1983; Ziegler
et al., 2010). The threshold value lies somewhere between 15 and 20 $\mathrm{W\,m^{-2}}$. Depending on the strength of additional factors
(such as GHGs or ice sheets) the threshold could be closer to the lower or the upper boundary of this range. Below 15 $\mathrm{W\,m^{-2}}$
until about 10 $\mathrm{W\,m^{-2}}$ the model simulates only a weak vegetation response that cannot develop into an AHP, probably because
within this range the amplifying feedback between vegetation and precipitation is not yet effective. Rossignol-Strick (1983)
found that the threshold for occurrence of sapropel layers in sediment cores from the Mediterranean Sea (an indication of past
AHPs) is about 19.8 $\mathrm{W\,m^{-2}}$ (41 $\mathrm{Langley\,day^{-1}}$). How close this value is from the estimate from CLIMBER-2 is useful as
additional validation for the model, even though we simulate Saharan greening and not sapropel formation.

From the control simulation we also find that the speed at which the AHP response occurs correlates well with the speed
of change in the monsoon index. This is valid during AHP onset and termination. However for the fastest growth rate of
vegetation to occur, it is important that the trends of change in the ice sheets and the GHGs radiative forcing are also favourable.
For instance, at AHP onset during MIS 5e the three forcing factors are synchronised with increasing monsoon index, strong
reduction of ice sheets and an increasing rate of warming from GHGs radiative forcing. Another example occurs at onset during
the Holocene AHP, when a modest increasing rate in the monsoon index is compensated by simultaneous negative trend in ice
sheets and warming trend from GHGs. Synchronisation of the forcings can explain why the speed of the modelled AHP onset
is generally faster than the subsequent AHP termination. This was also described for the Holocene AHP by Shanahan et al.
(2015). Because of a lag in the ice sheets rate of change with respect to the monsoon index change, and because of the erratic
rate of change in the GHGs, the three forcing factors are not usually in sync during AHPs terminations and the rate of change is
generally slower than their onset. The only AHP termination that is faster than its onset occurs in stage MIS 5a, when a strong
increase in ice sheets happens closely after the monsoon index negative peak rate of change. In Lézine et al. (2011) the role
of groundwater is described to explain why there is a rapid response during AHP onset and a subsequent slower more gradual
AHP termination (with aquifers providing base flow). However, the model's hydrology does not include this feature and we
focus only on the speeds of the forcings and their synchronisation.



We complement the study of the control simulation with a factor separation analysis. Its outcome confirms the dominant role of the orbital forcing in setting the amplitude of the modelled precipitation and vegetation responses of past AHPs. This was already described, for instance, in Tuenter et al. (2003), Kutzbach et al. (2020) and Blanchet et al. (2021). In particular, a factor

analysis in Blanchet et al. (2021) showed that the individual roles of the GHGs radiative forcing and the global ice volume are of secondary importance for the strength of simulated AHPs. What we do with the factor separation method is quantitative estimates of said dominance or secondary importance of these forcings. From our results, when we add together the individual contributions from GHGs and ice sheets to the AHP precipitation or vegetation coverage responses, we find they amount to less than 20 % of the change induced by the orbital forcing alone. In fact, according to the simulations the orbital forcing

alone (without effects from GHGs and ice sheets) should account for about 60 % (80 %) of changes in the AHP precipitation (vegetation) during the last 190 kyr. If its synergies are included, then the orbital forcing is responsible for more than 90 % of precipitation and vegetation changes. The factor separation also shows that GHGs and ice sheets have a larger impact on precipitation (and vegetation extent) when in synergy with the orbital forcing than individually. For near-surface temperature, in contrast, all the three forcing factors have a large effect individually and the synergies are all small in comparison.

After analysing the influence of the forcing factors on past AHP responses, we then evaluate the potential for future occurrences of humid periods. During the next 100 kyr AP the effects of changes in GHGs radiative forcing become larger while those of the orbital forcing become weaker. The future of the ice sheets is more uncertain, but a glacial inception is not likely to happen (Ganopolski et al., 2016) and if we ignore potential catastrophic consequences of a complete deglaciation, the impact of the ice sheets on AHPs should not change much from its current interglacial-like effect. Claussen et al. (2003) and D'Agostino

et al. (2019) already studied AHPs under future climate change and showed how a dynamically orbitally driven AHP response differs from a thermodynamically GHGs-driven AHP. Therefore we do not focus on the mechanisms that explain such differences but on the relative importance of each forcing during the AHP response. The main finding is that the GHGs radiative forcing has an amplification effect on the simulated vegetation response that is large enough to overcome the limiting threshold for Sahara greening imposed by the orbital forcing. An alternative interpretation is that the GHGs forcing lowers the tropical

insolation requirement for AHP onset. Here it is important to mention that we are neglecting biogeochemical effects, since we prescribe GHGs levels, therefore potential effects of feedbacks between the carbon cycle and vegetation are being ignored. The AHP responses are still paced by precession variations but the increased atmospheric moisture (product of GHGs radiative warming) compensates for a weaker gradient in insolation.

## 5 Conclusions

We have used the climate model of intermediate complexity CLIMBER-2 to assess the role on AHP strength of three climate forcing factors: Earth's orbital parameters, GHGs radiative forcing and ice sheets extent. Consistent with previous studies we find the model simulates reasonably well the evolution of past AHPs in terms of timing and magnitude. A newly discovered feature is the model contains the threshold behaviour of AHPs associated with the orbital forcing previously found in proxy data. For the Saharan greening response during AHPs, the simulated threshold in the monsoon index of tropical insolation lies

within 15 and 20 $\mathrm{W\,m^{-2}}$. Besides this critical value, the monsoon index (proxy of orbital forcing) also correlates well with the simulated precipitation and vegetation responses during AHPs, not only with their magnitudes, but also with their rates of change: higher monsoon index values lead to higher magnitudes and faster changes. We show also that for fast changes to occur it is important that the forcings are synchronised. In the simulations this happens most often during AHP onset than termination and therefore the onset rate of change is generally faster.

In addition to the results from a control experiment we include two complete factor separation analyses that make explicit the dominant role of the orbital forcing setting the strength of past AHPs. The individual effects of the GHGs and ice sheets put together cause less than 20 % of the change induced by the orbital forcing. The effects of GHGs and ice sheets are notably greater when in synergy with the orbital forcing than separately. When we add together the effects of the orbital forcing and its synergies it amounts to over 90 % of the change in the AHP precipitation and vegetation responses. Despite the dominance

of the orbital forcing, when we extend the simulations to cover the next 100 kyr AP we find that increased GHGs radiative forcing lowers the critical value of monsoon index that must be crossed to simulate Saharan greening and the subsequent AHP response.

These findings both support previous research and contribute new insights to our understanding of AHPs dynamics. In several cases we are able to extrapolate previous knowledge from the Holocene AHP to earlier humid periods. In general the

results agree with the consensus on predominance of the orbital forcing and make explicit the minor role that additional forcing from GHGs and ice sheets played during previous AHPs. However, because we consider both past and future simulations of AHPs we are able to show that GHGs may be more important than previously thought. A key message from this work is that even though the orbital forcing is the leading factor setting intensity and timing of AHPs, the atmospheric levels of GHGs have the potential to lower the insolation requirement for AHP onset. Such an effect of GHGs in the past might have been hidden

by the strength of the non-linear response to the orbital forcing, combined with a relatively narrow range of GHGs variability, yet it may be particularly important when considering future occurrences of AHPs under climate change scenarios.

*Code and data availability.* Model source code of CLIMBER-2 is available upon request. Data and post-processing Python scripts to reproduce the authors' work are archived by the Max Planck Institute for Meteorology and are accessible without any restrictions at http://hdl.handle.net/21.11116/0000-000A-1217-8. The Ehrmann and Schmiedl (2021) data in Fig. 2 are available at https://doi.org/10.1594/

PANGAEA.923491, last access: 13 May 2021. The Lisiecki and Raymo (2005) data shown in Fig. A1 are available at http://lorraine-lisiecki. com/LR04stack.txt, last access: 16 December 2020.

## Appendix A:  Forcings, monsoon index and MIS notation

Experiments with CLIMBER-2 differ only in the input data concerning the GHGs radiative forcing, the extent of ice sheets and orbital parameters (precession, obliquity and eccentricity). The GHGs radiative forcing input series agrees with data from

Antarctic ice cores (Petit et al., 1999; EPICA Community Members, 2004). It is described in Ganopolski et al. (2010) as an





**Table A1.** Orbital parameters in CLIMBER-2 simulations.

| Forcing type | Precession (longitude of perihelion) | Obliquity | Eccentricity | Monsoon index |
|---|---|---|---|---|
| Transient | Berger (1978) | Berger (1978) | Berger (1978) | Berger (1978) |
| Interglacial–like | 261.271° | 24.124° | 0.039 | 28.5 |
| Glacial–like | 97.771° | 22.787° | 0.019 | −4.2 |

equivalent $CO_2$ concentration ($C_e$) that includes variations of $CO_2$, $CH_4$ and $N_2O$. The total GHGs radiative forcing ($\Delta RF$) is computed as

$$\Delta RF = 5.35 \ln \frac{C_e}{C_0}, \tag{A1}$$

where $C_0$ is set to 280 ppm. The reference values for glacial and interglacial conditions are about 165 ppm and 280 ppm of
equivalent $CO_2$ respectively, which translate using Eq. A1 into the radiative forcing values summarised in Table 1.

In the case of ice sheets, Fig. 1 shows only a global cumulative sum. For our simulations the input for the model is a gridded time series that is itself model output from an earlier simulation by Ganopolski and Calov (2011), who ran CLIMBER-2 model employing the ice sheets model component SICOPOLIS (Greve, 1997). The ice sheets evolution is in agreement with the sea level changes data from Waelbroeck et al. (2002) and the ice sheets maps of Peltier (1994). The reference values in this case
are actually maps taken from the gridded series during the Eemian around 124 ka and the LGM around 21 ka. Their global cumulative ice volumes are those seen in Table 1. Sea level change corresponding to the Eemian ice sheets map is −0.21 m, while that of the LGM is −106.44 m.

CLIMBER-2 by default computes the orbital parameters using the formulae of Berger (1978). In the experiments with fixed orbital forcing, we switch off the module for the Berger (1978) computation, and fix the values of precession (i.e., angle from
autumnal equinox to perihelion), obliquity and eccentricity to those calculated for the Eemian and LGM periods. These values are shown in Table A1. To summarise them into a single value we use the monsoon index defined by Rossignol-Strick (1983). To compute the index it is necessary first to compute the cumulative insolation during the northern caloric summer season defined by Milankovitch (1941). We do this using the daily insolation values computed with the Berger (1978) theory and the equations presented in Vernekar (1972). The caloric summer insolation is obtained for latitudes at the North Tropic near
$23.45°$ N ($I_T$) and at the Equator ($I_E$) and then the monsoon index ($MI$) at millennia before the present-day time slice $t$ is

$$MI^t = 2I_T^t - I_E^t. \tag{A2}$$

Figure 1 shows in fact the variations in the monsoon index from the 1950 Common Era (CE) reference value of about $482 \, \mathrm{W \, m^{-2}}$ (around 995 $\mathrm{Langley \, day^{-1}}$). Because AHPs occur usually at peak times of this index, it is convenient for the discussion to be able to identify every monsoon index maximum. Therefore, to label them we use the MIS stage (or substage) name that occurred at the time of the monsoon index maximum. This is also useful because MIS names give hints about the





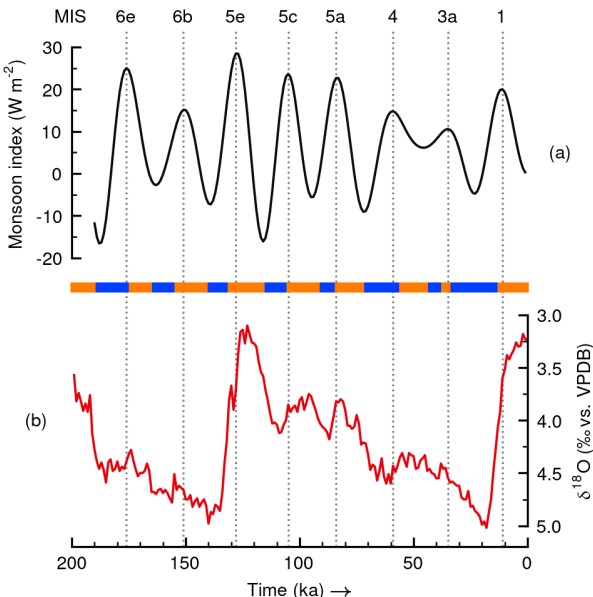

**Figure A1.** Notation for peaks of (a) monsoon index uses MIS labels on top, following the scheme in Railsback et al. (2015), who used the (b) Lisiecki and Raymo (2005) marine isotope stack data to label the time slices shown in a bar with different colours. The isotope data are with respect to the standard Vienna Peedee Belemnite (VPDB). Vertical lines indicate the monsoon index peaks.

state of the climate system at the time. In spite of there not being consensus in some of the names, here we make use of the notation scheme put forth by Railsback et al. (2015). Figure A1 shows the monsoon index in the control experiment and the Lisiecki and Raymo (2005) dataset used by Railsback et al. (2015). In Fig. A1 a thin bar with coloured boxes shows the different periods they identified. Vertical lines then help us choosing the names for each monsoon index peak.

**Appendix B: Equations of separation analyses**

We perform experiments following the separation method of Stein and Alpert (1993). In this case it is done for three factors: (1) GHGs radiative forcing, (2) global ice volume and (3) orbital parameters. Consequently $2^3$ simulations are needed for one separation analysis. The technique uses deviations of a simulation target with respect to a baseline or control state (experiment E0). We choose two different simulation targets: (1) global interglacial conditions with a strong AHP, and (2) global glacial

conditions with no AHPs. The targets correspond to simulations EI7 and EG7 respectively. Therefore we do two separation analysis for AHPs: one from an interglacial perspective and another from a glacial one. The equations for the separation



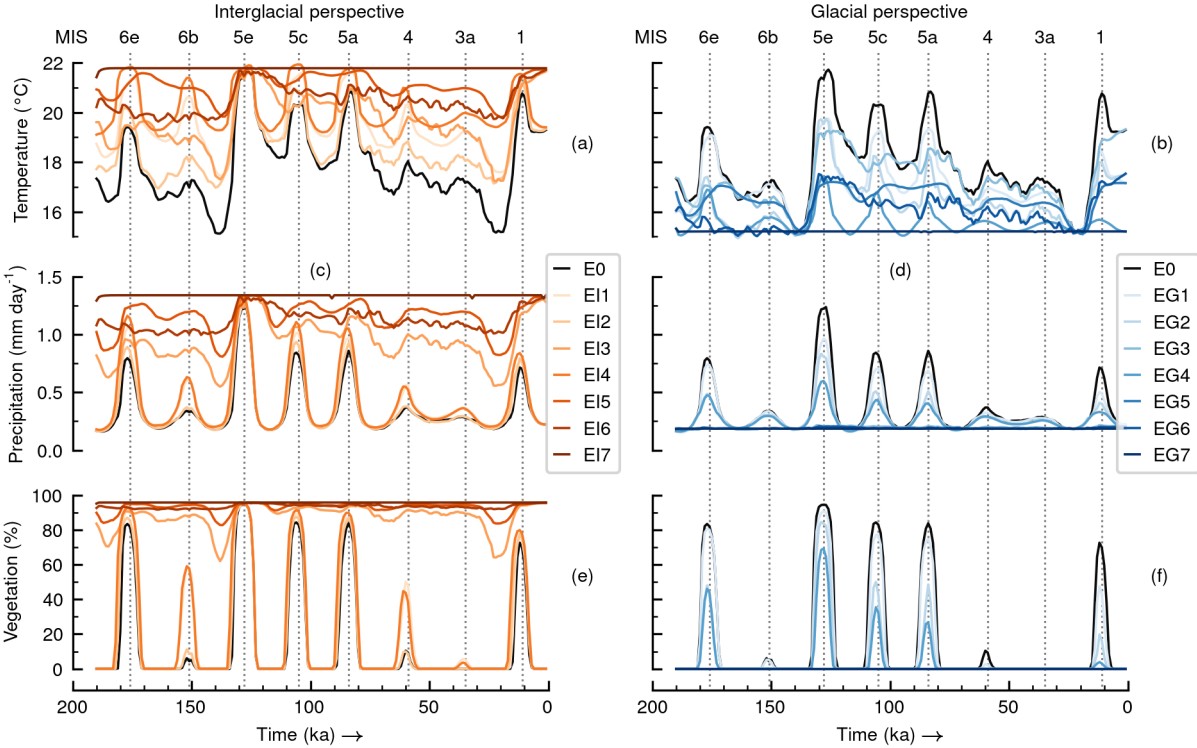

**Figure A2.** Results of annual means of (a, b) near–surface temperature, (c, d) daily precipitation and (e, f) vegetation fraction in the Sahara for all simulations involved in the separation analyses from interglacial and glacial perspectives. Vertical lines indicate monsoon index maxima and are labelled using MIS names on top.

analysis from the interglacial perspective are

$$\hat{f}_0 = f_0, \tag{B1}$$

$$\hat{f}_{\text{I, GHG}} = f_{\text{I, GHG}} - f_0, \tag{B2}$$

$$\hat{f}_{\text{I, Ice}} = f_{\text{I, Ice}} - f_0, \tag{B3}$$

$$\hat{f}_{\text{I, Orbital}} = f_{\text{I, Orbital}} - f_0, \tag{B4}$$

$$\hat{f}_{\text{I, GHG + Ice}} = f_{\text{I, GHG + Ice}} - (f_{\text{I, GHG}} + f_{\text{I, Ice}}) + f_0, \tag{B5}$$

$$\hat{f}_{\text{I, GHG + Orbital}} = f_{\text{I, GHG + Orbital}} - (f_{\text{I, GHG}} + f_{\text{I, Orbital}}) + f_0, \tag{B6}$$

$$\hat{f}_{\text{I, Ice + Orbital}} = f_{\text{I, Ice + Orbital}} - (f_{\text{I, Ice}} + f_{\text{I, Orbital}}) + f_0, \tag{B7}$$

$$\hat{f}_{\text{I, GHG + Ice + Orbital}} = f_{\text{I, GHG + Ice + Orbital}} - (f_{\text{I, GHG + Ice}} + f_{\text{I, GHG + Orbital}} + f_{\text{I, Ice + Orbital}}) + (f_{\text{I, GHG}} + f_{\text{I, Ice}} + f_{\text{I, Orbital}}) - f_0, \tag{B8}$$

where letters with hats in the left-hand side are the estimated effects on the simulation outcome of each forcing factor alone when having a single subscript or together in synergy with others when multiple subscripts. The letters on the right-hand side





(known as climatic fields) represent the model output for a variable taken from the set of experiments part of the analysis. Therefore what is shown in Fig. 2 are the fields $f_0$ from simulation E0 for temperature, precipitation and vegetation. Likewise $f_{\text{I, GHG + Ice + Orbital}}$ represents the variables when taken from simulation EI7. All fields for these variables are shown in Fig. A2, including all the other simulations. For the separation analysis from the glacial perspective the equations are

$$\hat{f}_0 = f_0, \tag{B9}$$

$$\hat{f}_{\text{G, GHG}} = f_{\text{G, GHG}} - f_0, \tag{B10}$$

$$\hat{f}_{\text{G, Ice}} = f_{\text{G, Ice}} - f_0, \tag{B11}$$

$$\hat{f}_{\text{G, Orbital}} = f_{\text{G, Orbital}} - f_0, \tag{B12}$$

$$\hat{f}_{\text{G, GHG + Ice}} = f_{\text{G, GHG + Ice}} - (f_{\text{G, GHG}} + f_{\text{G, Ice}}) + f_0, \tag{B13}$$

$$\hat{f}_{\text{G, GHG + Orbital}} = f_{\text{G, GHG + Orbital}} - (f_{\text{G, GHG}} + f_{\text{G, Orbital}}) + f_0, \tag{B14}$$

$$\hat{f}_{\text{G, Ice + Orbital}} = f_{\text{G, Ice + Orbital}} - (f_{\text{G, Ice}} + f_{\text{G, Orbital}}) + f_0, \tag{B15}$$

$$\hat{f}_{\text{G, GHG + Ice + Orbital}} = f_{\text{G, GHG + Ice + Orbital}} - (f_{\text{G, GHG + Ice}} + f_{\text{G, GHG + Orbital}} + f_{\text{G, Ice + Orbital}}) + (f_{\text{G, GHG}} + f_{\text{G, Ice}} + f_{\text{G, Orbital}}) - f_0, \tag{B16}$$

where now, for instance, there is $f_{\text{G, GHG + Ice + Orbital}}$ to represent the variables taken from simulation EG7. We do a total of 15 simulations (and not 16) for the two separation analyses because they share the same baseline state E0.

## Appendix C: Glacial separation analysis

The factor separation analysis from the glacial perspective is shown in Fig. C1. In this case deviations in EG7 from the control experiment E0 are negative. In simulation EG7 there is a permanent non-AHP state. Different to the interglacial case, the effect of the ice sheets is larger than that of the GHGs or the orbital forcing in some cases. Saharan temperature negative changes in Fig. C1a happen everywhere except during glaciations before the Eemian interglacial around $140$ ka and the LGM. Average contributions to temperature change during the series at times close to peaks of monsoon index in Fig. C1a (when changes are most visible) are $-1.0\,°\text{C}$ from GHGs ($-25.8\,\%$ of change from E0), $-2.0\,°\text{C}$ ($-49.4\,\%$ of change from E0) from ice sheets and $-1.5\,°\text{C}$ from orbital forcing ($-38.6\,\%$ of change from E0). There is a rather small counteracting (positive) temperature change from the synergy between ice sheets and orbital forcing that amounts to about $0.5\,°\text{C}$ ($+13.1\,\%$ of change from E0).

Glacial precipitation (Fig. C1b) and vegetation (Fig. C1c) reductions only occur at peak times of the monsoon index (where AHPs occur in control E0). For these two variables the deviations from the control experiment are bounded at $0\,\text{mm}\,\text{day}^{-1}$ and $0\,\%$. Due to these bounds the synergies are not reliable and they show in Fig. C1 as the symmetrical opposite shapes of their individual contributions for GHGs and ice sheets. Average contributions to precipitation reductions near monsoon index peaks in Fig. C1b (not the whole series) are about $-0.1\,\text{mm}\,\text{day}^{-1}$ from GHGs ($-13.0\,\%$ of change from E0), $-0.2\,\text{mm}\,\text{day}^{-1}$ from ice sheets ($-27.0\,\%$ of change from E0) and $-0.4\,\text{mm}\,\text{day}^{-1}$ from orbital forcing ($-54.0\,\%$ of change from E0). For vegetation change near times of peak monsoon index in Fig. C1c (not the whole series) contributions average about $-7.3\,\%$ from GHGs



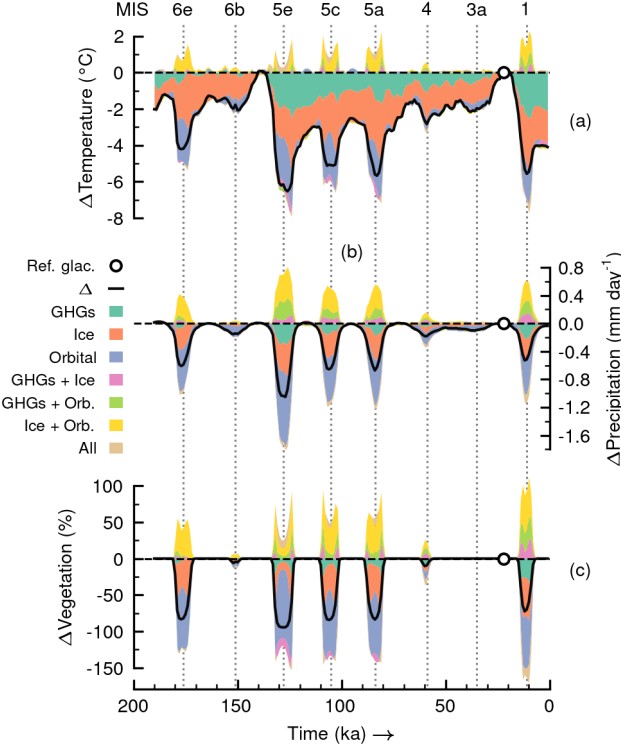

**Figure C1.** Factor separation analysis in Sahara from glacial perspective for annual means of (a) near-surface temperature, (b) daily precipitation and (c) vegetation fraction. Colours indicate contributions from individual forcings and related synergies to the total deviation in EG7 (glacial non-AHP) from the control experiment ($\Delta = $ EG7 $-$ E0). A marker shows the location of the reference glacial state, where the difference between EG7 and E0 is minimum. Vertical lines indicate monsoon index maxima and are labelled using MIS names on top.

($-8.3$ % of change from E0), $-26.0$ % from ice sheets ($-29.0$ % of change from E0) and $-49.0$ % from orbital forcing (55.0 % of change from E0). We reach a similar conclusion as from the interglacial perspective: contributions to temperature
are proportional to the size of the change in the forcings, while for precipitation and temperature the orbital forcing changes are the main cause of change.

*Author contributions.* MC, VB and MDV designed the research idea. TK and VB contributed to the experimental design and provided model code and input data. MDV performed the model experiments. All authors contributed to the analysis of results and to the preparation of the manuscript.

*Competing interests.* At least one of the (co-)authors is a member of the editorial board of *Climate of the Past*. The peer-review process was guided by an independent editor and the authors have no other competing interests to declare.



*Acknowledgements.* We thank Roberta D'Agostino (MPI-M) for helpful comments on an earlier version of the manuscript. Gerhard Schmiedl (UHH), Jürgen Böhner (UHH) and Jürgen Bader (MPI-M) provided valuable input and discussions. Jochem Marotzke (MPI-M), Dallas Murphy and participants of "S_41 Advanced Scientific Writing" (2021) also helped improve the writing in the manuscript. This work contributes to the project "African and Asian Monsoon Margins" of the Cluster of Excellence EXC 2037: Climate, Climatic Change, and Society (CLICCS). We acknowledge support of the German Climate Computing Center (DKRZ) in providing computing resources and assistance. The article processing charges for this publication were covered by the Max Planck Society. Data analysis and figures were produced using Python, including libraries NumPy, Matplotlib, xarray, SciPy, pandas, statsmodels, seaborn, sdt-python and cftime.






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
