# Peer review of "Threshold in orbital forcing for Saharan greening lowers with rising levels of greenhouse gases"

_Climate of the Past, 2022_

## Author Response (AR1)

**Author's Response on CP-2022-26**

Mateo Duque-Villegas (on behalf of all co-authors)
*mateo.duque@mpimet.mpg.de*

1st July 2022

We would like to thank all reviewers, editor Ran Feng and the Copernicus Editorial Team for their help in improving our manuscript. Below we compile all reviewers' comments in blue, our responses in black and **manuscript modifications in bold red**. In the modifications we refer to the line numbers in the revised manuscript.

**Community Comment 1 (CC1) by Zhengyu Liu**

We thank Dr. Zhengyu Liu very much for carefully reading our manuscript and for providing constructive remarks.

[0] This paper discusses the simulation of the North Africa monsoon and vegetation in the last 190,000 years. In particular, it highlights that an increased GHG lowers the threshold for Africa Humid Period (AHP) in the vegetation coverage. The paper is interesting and should be published. But, the paper would be more interesting to readers if some points can be clarified before publication.

**Major questions**

[1] The first question is on the mechanism of this threshold change in the model. Why is the threshold reduced (instead of increased) at a higher CO2? Can some specific sensitivity experiment be performed to show this change of threshold is caused by some vegetation (model) property/threshold, changing at different levels of CO2?

Changes in orbital forcing and GHGs radiative forcing have an amplifying effect on the simulated climate. Hence, with higher GHGs levels the system responds earlier to smaller orbital forcing. This has been demonstrated by, for instance, sensitivity experiments in Claussen et al. (2003), who used the same model. They show how $CO_2$ and orbital forcings affect dynamic and thermodynamic contributions to Saharan precipitation. D'Agostino et al. (2019) also looked into these contributions in CMIP5 experiments and arrived at a similar outcome. Following Brovkin et al. (1997), we see that there is a minimum of Saharan vegetation (or precipitation) after which the precipitation–vegetation feedback sets in. With dynamic (circulation) effects alone, caused mainly by the changes in orbital parameters, the minimum triggering value of vegetation (or precipitation) is only reached when the insolation is strong enough (expressed in terms of a monsoon index - which will be changed, see below). However, when thermodynamical effects become stronger (increased GHGs, increased atmosphere warming, increased water vapour), the minimum value of vegetation (or precipitation) can be reached sooner or with a lower value of tropical insolation. Also see our response below to comment [4]. In a revised version of our manuscript we will update Line 361 to include a detailed explanation of this feedback mechanism, instead of only referring to previous studies.

**We added a new paragraph in the Discussion section at lines 386–400 that includes part of this explanation.**

[Figure]

Figure C5: Reprinted from Claussen et al. (2003): Saharan vegetation fraction (a) and annual mean precipitation Ps (in mm/day) (b) as function of model years for different scenarios of changes in atmospheric $CO_2$ concentrations. The thin curves in (b) refer to results of the atmosphere-ocean-only model, the thick curves to results of the fully coupled model.

[2] The second question is on the role of vegetation feedback. Does this model has a positive vegetation feedback on precipitation in N. Africa? Or What is the role of vegetation feedback here? It seems to me in Fig.3 that the threshold is present only for vegetation, not for precipitation. If vegetation has a strong positive feedback on precipitation, I would also expect a threshold appearing on precipitation.

Yes, the model has a positive vegetation feedback on precipitation in North Africa. This has been clearly demonstrated in Claussen et al. (2003). Figure 5 in this paper – which we reprint here for your convenience as Fig. C5 – shows sensitivity experiments with dynamic vegetation switched on (thick lines in Fig. C5b) and off (thin lines in Fig. C5b). From Fig. C5, one could compute the ratio of an increase in precipitation with vegetation dynamics switched on ($\Delta P(V)$) and with vegetation dynamics switched off ($\Delta P(0)$), and one would arrive at a almost linear increase of $\Delta P(V) / \Delta P(0)$ with vegetation fraction f.

Regarding the threshold behavior in precipitation, if one focuses on the summer (orange-yellow line) in Fig. 3a, a case could be made that the threshold also exists for precipitation, since there is a jump from 1 to 2 mm day$^{-1}$. In fact, if we lower a penalty parameter in the changepoint analysis function, a threshold can also be detected for precipitation. For simplicity, we focus only on the one abrupt change that seems the least sensitive to the changepoint method. In a revised version of our manuscript we will expand on Section 2.1 our description of VECODE – the dynamical vegetation component of the model. Also in Section 3.2 we will explain how the changepoint method did not find immediately an abrupt jump for precipitation.

**We added a new paragraph in Section 2.1 in lines 77–83 with a description of the dynamic vegetation model. We explain in Section 3.2 in lines 201–202 how the change-point analysis does not find a change-point in precipitation. We added a new sentence in lines**

**347–348 pointing to the sensitivity experiments of Claussen et al. (2003).**

[3] Related to this, the forcing factor separation shows a big difference between precipitation and vegetation, with orbital forcing dominant on vegetation, but not on precipitation.

We are afraid there is a misunderstanding, but we do not think Fig. 6 shows a big difference between vegetation and precipitation, since in both of them the orbital forcing is the dominating factor (both have a lot of pale blue). It is true there is a difference, which is related to the synergistic contributions (yellow and green) of GHGs and ice sheets with the orbital forcing. The difference could be explained by the fact that vegetation has an upper boundary at 100 %, and therefore the synergistic contributions cannot be as effective as in the precipitation response, since the orbital forcing alone already causes most of the changes to reach almost 100 %.

**Add note about substantial blue colour in line 263.**

[4] It may be interesting to perform an experiment with the vegetation fixed to see how the precipitation changes. Even only one section of the simulations over 1-2 AHPs will be interesting.

We agree with the reviewer, and we will refer to the earlier CLIMBER-2 study by Claussen et al. (2003) as mentioned above.

**The new sentence in lines 347–348 refers to these sensitivity experiments.**

**Minor questions**

[5] The definition of monsoon index is confusing to me. It itself sounds like an index for the monsoon response, but, it is really the insolation forcing. Perhaps, it should be changed to Monsoon Forcing Index.

We used term Monsoon Index as it was defined and used in the classical paper by Rossignol-Strick (1983). But we agree with the reviewer and the review by Dr. Brierley (RC1) that for the readers' convenience, the term should be changed. In a revised version, we will refer to 'orbital forcing' or 'monsoon forcing index' as suggested.

**All instances of "monsoon index" were updated to "monsoon forcing index".**

[6] Why EI interglacial has a negative GHG of -2.8 W/m2? I thought interglacial has a higher CO2?

We are unable to find this in the text. In Table 1, EI experiments have 0.0 W m-2 as radiative forcing, since GHGs levels where close to an equivalent $CO_2$ concentration of 280 ppm.

**No action.**

[7] 3: Caption needs to be more specific. What is a dot for? Correlation thorough the entire period, or AHPs?

We agree with the reviewer that the caption needs to be expanded to briefly include the more detailed explanation given in the text (the dots refer to the values in Table 2).

**Caption of Fig. 3 now explains where the dots come from.**

[8] The title is on GHG lowers the threshold. But the paper discusses much beyond this, and actually, this point is somewhat lost in the discussion, at least, it does not read to me like the major point of the paper, because of so many other things discussed. Maybe this is indeed the most novel point, while other points are just consistency check...If that is the case, other parts can be simplified to highlight this novel point.

We agree with the reviewer that the title of our manuscript only refers to one highlight of our paper. We will ask the Editor, whether a change of title is possible. We think of "Effects of orbital forcing, greenhouse gases and ice sheets on Saharan greening in past and future multi-millennia" as a broader title.

**The title was updated to "Effects of orbital forcing, greenhouse gases and ice sheets on Saharan greening in past and future multi-millennia".**

**Referee Comment 1 (RC1) by Chris Brierley**

We thank Dr. Chris Brierley very much for carefully reading our manuscript and for the constructive remarks.

[0] This is a good paper that presents some interesting new simulations. I appreciate the work that's gone into these runs and their analysis and can readily see this manuscript being published in Climate of the Past. There are some aspects of it that need clarification before publication, and I think a little bit of further analysis would greatly enhance the reach of this manuscript. I especially appreciate the data and code placed in the online repository.

[1] The model description (Sect 2.1) mentions nothing about the land surface model. Given the importance of the vegetation fraction in this manuscript, you need to provide some information about how vegetation is simulated by the model (tree, grass etc) – and what, if any, feedbacks it has on the atmosphere.

We agree and in a revised version of the manuscript we will expand our description of the vegetation model component VECODE (see our reply to CC1).

**We added a new paragraph in Section 2.1 in lines 77–83 with a description of the dynamic vegetation model.**

[2] I feel the analysis about the rates of change (Sect 3.3) is out of place in this manuscript. It seems to invoke a fundamentally different conception of an AHP to the other work. The rest of the work talks about thresholds (implying transitions between bistable states). Yet this section discusses the speed of the changes as being related to the speed of forcing changes irrespective of their location w.r.t. the thresholds. Personally, I feel this aspect of the research should be removed to focus more on the subject in the title.

We understand the title does not fully reflect the breadth of our study. This also is a prevalent comment in all reviews. Therefore, we will change the title after consultation with the Editor. Regarding Section 3.3, we partially agree. We still think that the section is interesting, since we find a threshold in the changes of vegetation, but no threshold in the speed of this change. But instead of discussing these results in a separate section, in a revised version of our manuscript we will shorten the discussion and put it as extra paragraphs into Section 3.2.

**The title was updated to "Effects of orbital forcing, greenhouse gases and ice sheets on Saharan greening in past and future multi-millennia". The section about rates of change was summarised as a new paragraph in Section 3.2 in lines 213–226. We removed Table 3 and the original Fig. 5, which were related to the analysis of the rates of change.**

[3] You discuss the threshold as a function of the maximum orbital forcing. This may be appropriate for precipitation, but is this really the best way to think of vegetation threshold? Intuitively, I see a threshold as being lower than the maximum value with the intensity of the vegetation response driven by the time spent over that threshold.

Indeed, we focus on the threshold in the orbital forcing, following the classical paper by Rossignol-Strick (1983), who has found a threshold in monsoon forcing above which sapropels (proxy for AHPs) in marine cores from Eastern Mediterranean occur. We were excited to see that in our model, we capture the same threshold behavior as found in the proxy data. The threshold is a value (between 15–20 W m-2) lower than the maximum value of the orbital forcing (about 30 W m-2 in the last 200 ka), and the intensity of the vegetation response is driven by the time spent over the threshold and by how much the forcing exceeds the threshold.

**No action.**

[4] It is not clear precisely what is plotted in the trajectories of simulated data. Are these the data for a single grid box? If so, which one? Is the vegetation fraction presented a proportion of this grid box, with the rest of it being bare soil?

Indeed, in the coarse-scale model CLIMBER-2, the Sahara is represented as one grid box (see previous CLIMBER-2 publications). We will point out this more clearly in a revised version of our manuscript.

**We updated all figure captions to include the words "the Sahara grid box". In the paragraph in lines 77–83 we explain how the non-vegetated areas can be desert fraction.**

[5] Why have you selected only the past 190 kyr (Sect 2.2)? I presumed this was motivated by the 2 references cited on L36 – although you should make this explicit. It seems though that Ehrmann & Schmiedl review back to 200ka and Blanchet et al seems to go back to 160ka from their Fig 3. I don't expect you to redo any simulations – your start date is fine for the science. But it needs a solid motivation written in the paper.

Yes, we chose this time window based on the data sets in Ehrmann et al. (2017), Ehrmann et al. (2021). We will motivate our choice more clearly when revising our manuscript.

**Added sentence in lines 61–62 explaining that the simulation time allows comparison with Ehrmann et al. (2021) data.**

[6] There is no discussion in the paper of internal variability in the simulations. My own work (Brierley et al, 2018, https://www.nature.com/articles/s41467-018-06321-y) building of Zhengyu Liu's model relies quite heavily on the fact that the AHP transitions involved some stochasticity. I suspect this will be case for CLIMBER-2 as well, and that this would explain the difference in precipitation at MIS5e between EI7 and E0 in Fig6b. Again, I don't think any additional analysis is needed – just some discussion of its implication for your analyses.

CLIMBER-2 is a statistical–dynamical model that has by design no short-term weather variability, but only climate variability at time scale of decades and longer. The effect of weather on the climatic circulation (mainly the meridional heat and momentum transport) is parameterized (Petoukhov et al., 2000). Therefore, we use century-scale averages of the model output. The differences between EI7 and E0 at MIS5e are actually due to the combination of prescribed forcing factors, which all happen closely around 125 ka, but not exactly at the same time as in E0. To see this, one has to closely look at Fig. 1a and b (GHGs and ice sheets) and notice the small differences between the black solid and orange dashed lines around MIS5e. In a revised version of our manuscript we will discuss the internal variability of the statistical–dynamical model CLIMBER-2.

**A new opening paragraph in the Discusion in lines 320–328 includes part of this explanation. A new sentence in lines 250–251 explains the slight offset between E0 and EI7 during MIS 5e.**

[7] You could go further with your simulations and combine the results from the future simulations with that of EI2, EI4 and EI6 to perform an analysis similar to that in Fig. 3 to quantify the impact of GHG forcing on the orbital threshold. As currently written this feels like a missed opportunity to really demonstrate the statement in the title.

We welcome this suggestion and we will study the output of such analysis.

**We have roughly quantified the impact of the GHGs forcing on the orbital threshold. New sentences in lines 207–212 explain this finding for the past AHPs. We link this finding to the future simulations in lines 306–315. In lines 387–389 in the Discussion, and in the Conclusions in lines 439–441 we also refer to this finding.**

**Other comments**

[8] 'Synergical' feels very awkward – try 'synergistic'

This will be fixed in a revised version.

**Fixed in all instances.**

[9] I agree that with Dr Liu that a slight rebranding of the Monsoon Index would be helpful

This will be fixed in a revised version as explained in our response to Dr. Liu (CC1).

**All instances of "monsoon index" were updated to "monsoon forcing index".**

[10] You should explain how the lagged peaks in Fig 2a reflect the intensity during the sapropel. You make no comment about the split event at 5c in SL77. Why are these better measures of intensity than something like the co-eval Ba/Al ratios measured by Zeigler et al (2010)?

The interruption in sapropel S4 is a known feature of sapropels from the Eastern Mediterranean, and is probably related to postdepositional "burn down" events via redox reactions in the sediments (Emeis et al., 2003; Grant et al., 2016). We do not argue one proxy data to be better, however, the data from Ehrmann et al. (2021) is associated with weathering (rainfall) and accumulation of minerals in water bodies across (most likely) North Africa, while Ba/Al is a measure of seabed primary productivity susceptible to multiple oceanic processes also discussed by Ziegler et al. (2010). Though our study is a purely modelling effort, we do agree a bit more context about the data could be helpful. In a revised version of our manuscript we will add extra remarks about proxy data in Section 3.1.

**In Section 3.1 in the paragraph in lines 151–166 we added part of this explanation, also including the additional proxy data suggested by another referee.**

[11] "reckon" on L169 sounds informal. Please replace.

This will be fixed during revision.

**Fixed.**

[12] You are too precise stating that the change point at 20Wm-2. Surely all you can tell is that its between 15-20 Wm-2.

The precision comes from the monsoon index value at MIS 1 (Holocene) in Table 2, which is 20.0 W m-2 (i.e., the method does not compute new numbers). The changepoint method is only selecting the value in Table 2 (column 5) where there is a jump in data. In a revised version we will add a parenthetical note about this "20 W m-2" being the Holocene value.

**The change-point method finding is now presented more clearly in lines 199–200. In the same lines we now speak of a threshold range between 15-20 W m-2.**

[13] How do you justify LOWESS smoothing all the forcing in Fig. 4, but not the simulated vegetation fraction? [I recommended cutting this section above]

The smoothing is only applied to the GHGs and ice sheets series to ease the visual inspection of trends. We will remove smoothing of forcings in a revised version of the manuscript.

**The figure was re-plotted without smoothing.**

[14] I strongly suspect that the analysis in Fig 5 would have also show the rates of initiation and termination of the AHP events is strongly correlated to the peak monsoon index. How can be sure that your style of analysis is more appropriate. [I recommended cutting this section above]

We thank the reviewer for this remark. Please see our response here to comment [2].

**The figure was removed in the process of merging this part into Section 3.2. However, a sentence in lines 218–219 refers to this finding.**

[15] 6. I like this figure, but can you please check that it works for color-blind individuals.

This was checked with https://www.color-blindness.com/coblis-color-blindness-simulator/ and the figure works for some but not all types of colour blindness. We will test for better colour sets and update the colours in Fig. 6 during the revision.

**The figure was re-plotted with colorblind-safe colours and including hatching.**

[16] This sentence seems odd. If you really feel that it is only the weak orbit that matters, then please rephrase to avoid the conflation with 'glacial times' – as that phrasing intuitively suggest that GHG and ice-sheets play a role. You might want to try: "This analysis demonstrates that it is the relatively low maximums in orbital forcing that result in the absence of AHP conditions at 6b, 4 and 3a – rather than the low GHG forcing or large ice sheets."

Assuming this statement refers to Line 274, we welcome this suggestion. We will re-phrase the sentence in Line 274 in a revised version of our manuscript.

**Fixed exactly as suggested in lines 288–290.**

[17] It would be instructive to take the work about future AHP conditions a little further. Can you find a way to quantify the impact of GHG forcing on the orbital threshold. I feel that there should be enough data here.

We appreciate this suggestion and we will assess possible ways to achieve this.

**We have roughly quantified the impact of the GHGs forcing on the orbital threshold. New sentences in lines 207–212 explain this finding for the past AHPs. We link this finding to the future simulations in lines 306–315. In lines 387–389 in the Discussion, and in the Conclusions in lines 439–441 we also refer to this finding.**

[18] I also wonder if you could provide some additional context for the future simulations for those of us not fully versed with the future carbon cycle pulses. As well as the GHG forcing, it might be helpful to plot global mean temperatures and atmospheric CO2 levels. In effect, I am wondering how the future AHP at M1 relates to proposed warming levels and safe operating spaces.

In a revised version we will expand details about future climate change scenarios in Section 2.3, and include global average temperature time series in Fig. 7.

**In Section 2.3, now lines 131–135 give a bit more context about the scenarios. Figure 6 (originally 7) now includes a panel with the global mean temperature and we referred to it in line 299.**

**Referee Comment 2 (RC2) by anonymous referee**

We thank the anonymous referee very much for carefully reading our manuscript and for providing constructive remarks.

[0] This study presents modeling results of the last 190,000 years of African rainfall and vegetation history, classifying certain thresholds as "African humid periods" and commenting on the strength, duration, and rates of change of these differing AHPs. The authors find that orbital forcing is the primary driver for changes in rainfall and vegetation extent during past AHPs, but that the sensitivity threshold of AHPs to orbital forcing is modulated by GHG concentrations. Future modeling experiments are also conducted that show future AHPs are more likely to occur with higher concentrations of GHGs, as future orbital insolation thresholds are too low to induce AHPs without GHG increases.

[1] This study is well motivated and provides novel findings with respect to previously unknown factors contributing to the strength, duration, and rates of change of past AHP. This paper is exceptionally well-written and clearly presents its results and conclusions. In addition to a few minor comments, I believe one area for improvement can come from some added discussion on the uncertainties present within the very coarse model resolution of CLIMBER-2. It is important to show that the authors have considered all of the uncertainties involved in using this specific model and conclude that these uncertainties do not impact the conclusions of this paper – i.e., this model is the perfect fit for use with this specific research question. I recommend this paper be accepted with some very minor revisions. I list each comment for the revised manuscript below.

We agree it is important to consider possible sources of uncertainty in our study. CLIMBER-2 model uncertainty has been previously found to be comparable to that of CMIP5 models on the global scale (e.g., Ganopolski et al., 2016). And the data we use to prescribe the forcing in the model is based on widely recognised and discussed data sets. Moreover, our results are obtained with an ensemble totalling 21 experiments. Therefore, we believe uncertainties in model and forcing data should not compromise our findings. In a revised version of the manuscript we will refer in Section 2 (model description) to the uncertainties in the model and in the forcing data.

**That the model is the perfect fit for this research question is reiterated in the new paragraph in the Discussion in lines 320–328. We mentioned how the model leaves out variability at shorter than decadal timescales.**

**Comments**

[2] It will strengthen the manuscript to elaborate upon the scale of the research question with regard to these simulations (for example: this study examines shifts in state of climate, such as desert vs. >50% vegetation cover, present within the single North Africa grid cell and does not require finer details with regard to the simulated climate) and how examination at this scale minimizes the large uncertainties present with using CLIMBER-2 to simulate paleoclimate. Bringing in discussion of multiple climate equilibria (green vs. desert) in northern Africa may help to strengthen this argument.

We welcome this suggestion and we will include in a revised version of Section 4 (discussion) a paragraph about the scale of our research question and about how the uncertainties in model and forcing data should not invalidate our discussion. At our scale of interest the possibility of multiple climate equilibria in North Africa has been discussed extensively.

**We modified the opening paragraph of the Discussion (lines 320–328) to explain that CLIMBER-2 is a statistical–dynamical model and that CLIMBER-2 has been successfully applied to a number of palaeoclimate studies. We mentioned how the model leaves out variability at shorter than decadal timescales.**

[3] In Table 1, it would be more clear to list "Monsoon index via orbital parameters" (or something like this) so to not confuse readers over what is being prescribed in the model. The authors prescribe orbital parameters, which in turn dictate the monsoon index, rather than directly prescribing "monsoon index" as a specific boundary condition. Slight added nuance to reflect this would preclude confusion for future readers.

This will be fixed in a revised version as explained in our response to Dr. Liu (CC1).

**All instances of "monsoon index" were updated to "monsoon forcing index".**

[4] In Table 1, what does GHG radiative forcing = 0.0 W/m2 correspond to? The base value is listed for monsoon index (line 428), so it would be helpful to include the same for GHG radiative forcing. Or if this value is more difficult to assess, at least define more clearly that deltaRF is a change from the modern day... which is what time period? 1950 CE?

Indeed we were wrongly saying "radiative forcing" when it should be "radiative forcing change" with respect to a preindustrial base concentration for $CO_2$ of 280 ppm ($C_0$). This is explained in Appendix A, but not in Table 1 or the main text. This will be fixed in a revised version of the manuscript.

**In all captions and the main text we now explain that the GHGs radiative forcing changes are with respect to a preindustrial concentration of equivalent $CO_2$ of 280 ppm.**

[5] On line 158, there are several studies with updated simulations using sophisticated models that could be cited here, in addition to Harrison et al. (2015). I would suggest adding at least a few of the following citations: Pausata et al. (2016), 10.1016/j.epsl.2015.11.049 Thompson et al. (2019), 10.1029/2018GL081225 Hopcroft et al. (2021), 10.1073/pnas.2108783118 Chandan & Peltier (2020), 10.1029/2020GL088728 Dallmeyer et al. (2020), 10.5194/cp-16-117-2020

Some of these were already mentioned in the introduction in Line 35. In the revision we will add "e.g." to this citation and include some of the others the reviewer suggests.

**Additional references were added to lines 35 and 177.**

[6] Both the interglacial and glacial factor separation analyses are important results of this paper, yet only one is presented in the main text. I would suggest the authors bring the glacial factor separation analysis into the main text as an additional figure. Or the authors could at least describe why they believe the interglacial case is more important than the glacial case and use this explanation to justify why the interglacial case is included in the main text while the glacial case is not.

We agree the glacial factor separation is a relevant complement to our method, and that is why it is included in our study. However, we think it does not add anything significantly new to our discussion and that is why it is in Appendix B. We will justify this decision in the main text around Line 271 in a revised version of the manuscript.

**The opening line of the paragraph in lines 285–291 now explains this decision.**

**Referee Comment 3 (RC3) by anonymous referee**

We thank the anonymous reviewer very much for carefully reading our manuscript and for providing constructive remarks.

[0] This paper is a valuable contribution to the theory underlying African Humid Periods and their variable forcings. The authors present a carefully considered set of intermediate complexity model simulations that allow for factor separation analysis. They have clearly produced a lot of data/results, and I appreciate the efforts they have made to condense the work to the most important points. I think the paper is close to being ready for publication. Here, I touch on some previous points by other reviewers that I agree with, and I add a couple of additional, minor points.

[1] The most substantial point that I wish to emphasize comes from Dr. Liu about how the main text is somewhat disconnected from the title. This is a substantial point only in that I think the paper could benefit from re-structuring the arguments, but I don't see this as necessary for publication. Specifically, I suggest re-framing the paper more explicitly as a comparative analysis of past and future AHPs. This would involve discussing the future simulations more prominently and, as other reviewers mentioned, diving into more detailed hypotheses as to how/why GHGs lower the orbital threshold. The question of whether emissions can compensate for low future eccentricity to cause future AHPs is thought-provoking, and the results—casting emissions scenarios as the primary determinant of the frequency and amplitude of future AHPs—could motivate much further research into GHG and orbital "synergies".

Please see our responses to Dr. Liu (CC1) and Dr. Brierley (RC1) on this topic. In a revised version we will re-structure parts of the text, expand on the model description and on the possible mechanisms for GHGs lowering the orbital threshold. We will also, in consultation with the Editor, consider updating the title if possible.

**The title was updated to "Effects of orbital forcing, greenhouse gases and ice sheets on Saharan greening in past and future multi-millennia". The section about rates of change was summarised as a new paragraph in Section 3.2 in lines 213–226. We removed Table 3 and the original Fig. 5, which were related to the analysis of the rates of change. We added a new paragraph in the Discussion section at lines 386–400 that includes an explanation of the threshold-lowering mechanism. We have roughly quantified the impact of the GHGs forcing on the orbital threshold. New sentences in lines 207–212 explain this finding for the past AHPs. We link this finding to the future simulations in lines 306–315. In lines 387–389 in the Discussion, and in the Conclusions in lines 439–441 we also refer to this finding.**

[2] A couple of smaller points that I agree with from other reviewers. I like Dr. Liu's suggestion to call "Monsoon Index" the "Monsoon Forcing Index". I also agree with Dr. Brierly that more background on the land surface model is needed, especially with respect to the threshold behavior, relevant feedbacks (including fire), and whether there is hysteresis. I also agree that the rate of change analysis could be removed. It's not currently grounded by anything in the discussion, and I agree that it is difficult to square with the threshold behavior.

Please see our response to Dr. Brierley (RC1) on these issues. In a revised version of the manuscript the monsoon index will be "re-branded", the model description extended and the rates-of-change part will be shortened and moved.

**All instances of "monsoon index" were updated to "monsoon forcing index".**

[3] My two suggestions are (1) to cite/discuss some more proxy work; and (2) be more explicit about any assumptions associated with factor separation analysis:

[3.1] The paper focuses on two records from the Mediterranean for comparison. However, there are other records that span the same time interval, and it is worth mentioning how they compare (amplitude, duration, etc) to the new model results. Two datasets that are particularly relevant are Miller et al. 2016 (JQS) and Skonieczny et al. 2019 (Sci. Adv). The Miller paper is useful for more directly comparing the vegetation results to a vegetation reconstruction; and the Skonieczny paper presents another dust flux record off West Africa.

We thank the reviewer for the literature suggestions and we will briefly discuss these additional proxy records in an updated version of the manuscript.

**In Section 3.1 in the paragraph in lines 151–166 we expanded the explanation of the proxy data, and included some of the additional proxy data suggested.**

> [3.2] One concern I have has to do with any assumptions inherent to the FSA (I don't have expertise in FSA, so please bear with me). It seems like one implicit assumption is that any non-linearities (when multiple forcings yield a different result than the sum of individual forcings) can only arise due to "synergies" or interactions between the forcings. That is, the response to any forcing is assumed linear so the responses can be summed together (and deviations from the sum are synergies). However, a non-linear response to a forcing (such as threshold behavior in vegetation %) could lead to "apparent synergies" between forcings that are actually projections of a non-linear response (rather than a non-linear interaction between forcings). For example, if GHGs and orbital forcing alone both cause a stepwise increase from desert-to-grassland, then FSA would expect the GHG + orbital simulation to produce this stepwise transition twice (without synergies) and any deviation from this would be attributable to synergies. The authors briefly touch on this specific case in line 265, but they relate the issue to the fact that vegetation % is bounded between zero and 100. However, maybe it's the case that the bounding only makes the issue clearly diagnosable. I'm curious if the issue is broader, applying to any non-linear response where the response is not necessarily the sum of its parts (even without "synergies"). I expect that a basic discussion of the assumptions of FSA would suffice here. If my comment about non-linear responses is entirely off-base, then maybe adding a sentence about why this intuition is wrong could be helpful for readers like me.

The factor separation analysis does not diagnose whether the response–factor relationship is linear or not. Instead, it diagnoses whether contributions to the response from multiple factors can be added linearly or not. It is possible that individual forcing factors cause non-linear responses, but they should always do it in a similar way, and therefore their joint effects could be added linearly to predict a full response. When they cannot be added linearly, then we need to include the synergies. In other words, synergies are "non-linear terms" in the sense that they show that individual effects from multiple factors cannot simply be added to predict the full response. The challenging part is assigning a physical meaning to the synergies that the separation method obtains. In a revised version we will expand on the description of the method to explain the synergies in this way.

**Line 114 now includes a short sentence about what the method assumes.**

**References**

Brovkin, V., Ganopolski, A., & Svirezhev, Y. (1997). A continuous climate-vegetation classification for use in climate-biosphere studies. *Ecological Modelling*, *101*(2–3), 251–261.

Claussen, M., Brovkin, V., Ganopolski, A., Kubatzki, C., & Petoukhov, V. (2003). Climate change in northern Africa: The past is not the future. *Climatic Change*, *57*(1-2), 99–118.

D'Agostino, R., Bader, J., Bordoni, S., Ferreira, D., & Jungclaus, J. (2019). Northern Hemisphere monsoon response to mid-Holocene orbital forcing and greenhouse gas-induced global warming. *Geophysical Research Letters*, *46*(3), 1591–1601.

Ehrmann, W., Schmiedl, G., Beuscher, S., & Krüger, S. (2017). Intensity of African humid periods estimated from Saharan dust fluxes. *PloS One*, *12*(1).

Ehrmann, W., & Schmiedl, G. (2021). Nature and dynamics of North African humid and dry periods during the last 200,000 years documented in the clay fraction of eastern mediterranean deep-sea sediments. *Quaternary Science Reviews*, *260*, 106925.

Emeis, K.-C., Schulz, H., Struck, U., Rossignol-Strick, M., Erlenkeuser, H., Howell, M. W., Kroon, D., Mackensen, A., Ishizuka, S., Oba, T., Sakamoto, T., & Koizumi, I. (2003). Eastern mediterranean surface water temperatures and $\delta 18o$ composition during deposition of sapropels in the late quaternary. *Paleoceanography*, *18*(1).

Ganopolski, A., Winkelmann, R., & Schellnhuber, H. J. (2016). Critical insolation–$CO_2$ relation for diagnosing past and future glacial inception. *Nature*, *529*(7585), 200–203.

Grant, K. M., Grimm, R., Mikolajewicz, U., Marino, G., Ziegler, M., & Rohling, E. J. (2016). The timing of mediterranean sapropel deposition relative to insolation, sea-level and african monsoon changes. *Quaternary Science Reviews*, *140*, 125–141.

Petoukhov, V., Ganopolski, A., Brovkin, V., Claussen, M., Eliseev, A., Kubatzki, C., & Rahmstorf, S. (2000). CLIMBER-2: A climate system model of intermediate complexity. Part I: Model description and performance for present climate. *Climate Dynamics*, *16*(1), 1–17.

Rossignol-Strick, M. (1983). African monsoons, an immediate climate response to orbital insolation. *Nature*, *304*(5921), 46–49.

Ziegler, M., Tuenter, E., & Lourens, L. J. (2010). The precession phase of the boreal summer monsoon as viewed from the eastern Mediterranean (ODP site 968). *Quaternary Science Reviews*, *29*(11-12), 1481–1490.